# Substitution Impact of Tuna By-Product Meal for Fish Meal in the Diets of Rockfish (*Sebastes schlegeli*) on Growth and Feed Availability

**DOI:** 10.3390/ani13223586

**Published:** 2023-11-20

**Authors:** Ran Li, Sung Hwoan Cho

**Affiliations:** 1Department of Convergence Study on the Ocean Science and Technology, Korea Maritime and Ocean University, Busan 49112, Republic of Korea; ytliran@126.com; 2Division of Convergence on Marine Science, Korea Maritime and Ocean University, Busan 49112, Republic of Korea

**Keywords:** fishmeal substitution, tuna by-product meal, economic profit index, rockfish (*Sebastes schlegeli*)

## Abstract

**Simple Summary:**

Fish meal is commonly used as the main protein source in aquafeeds. However, the increasing global demand and the overfishing of wild fish stocks commonly used as fish meal sources have restricted supply and increased the price of fish meal in recent decades. To achieve the goal of sustainable fish culture, fish meal replacement with an alternative source that is inexpensive and year-round available in aquafeeds is highly needed. Meanwhile, more than half of the total tuna is trashed as waste in tuna-canning processing plants. In the current study, we investigated the dietary substitution effect of fish meal with tuna by-product meal on growth, feed availability, and biochemical composition of the early stage of juvenile rockfish, one of the most widely farmed marine fish species in Eastern Asia. The results indicated that up to 40% of fish meal could be substituted with tuna by-product meal in the diets of rockfish without negatively influencing the growth, feed availability, and hematological parameters of rockfish. With the dietary replacement of 40% fish meal with tuna by-product meal in practical feeding, rockfish farmers are able to produce the highest economic return.

**Abstract:**

This experiment was performed to assess the substitution impact of fish meal (FM) with tuna by-product meal (TBM) in feeds on growth and feed availability of the early stage of juvenile rockfish (*S. schlegeli*). Six experimental feeds were prepared to be isonitrogenous and isolipidic. Fifty-five percent of FM was contained in the control (Con) diet. In the Con diet, 20, 40, 60, 80, and 100% FM were replaced by TBM, named the TBM_20_, TBM_40_, TBM_60_, TBM_80_, and TBM_100_ diets, respectively. A total of 540 early-stage juvenile rockfish averaging 2.4 g was divided into 18 tanks and hand-fed to satiation for 56 days. Weight gain and feed consumption of rockfish fed the TBM_20_ and TBM_40_ diets were comparable to rockfish fed the Con diet. The specific growth rate (SGR) of rockfish fed the Con diet was comparable to rockfish fed the TBM_20_, TBM_40_, and TBM_60_ diets. Feed efficiency, biometric indices, hematological parameters, proximate composition, and amino acid profiles of rockfish were not impacted by dietary treatments. The economic profit index (EPI) of the Con, TBM_20_, and TBM_40_ diets were higher than that of all other diets. FM up to 40% could be substitutable with TBM in the diets of rockfish without deteriorating weight gain and feed consumption, but producing the highest EPI.

## 1. Introduction

Ranking second high after olive flounder (*Paralichthys olivaceus*) in Korea, the aquaculture production of rockfish (*Sebastes schlegeli*) reached 16,189 metric tons in 2022 in Korea [1]. Rockfish are widely distributed along the coasts of Korea, China, and Japan [2]. However, as a carnivorous fish species, Asian farmers, especially Korean and Chinese farmers, prefer feeding rockfish on raw fish or raw fish-basal moist pellet (MP) rather than formulated pellet (FP), although this practical feeding is costly and inappropriate for long-term aquaculture practices [1,3,4]. Commercially available nutrition-balanced FP, on the other hand, could efficiently produce fish in a more environmentally friendly manner than MP [5]. According to Kim et al. [6], rockfish fed the MP achieved lower survival than fish fed on the extruded pellet in the 12-week feeding trial. 

Fish meal (FM) is commonly utilized as the main protein source in FP of carnivorous and omnivorous fish species because of its high protein content, balanced amino acid (AA) profiles, and good palatability [7,8]. However, the increasing global demand and the overfishing of wild fish stocks have restricted supply and increased the price of FM in recent decades [9]. Therefore, several studies were carried out to explore the alternatives of FM in fish feeds, including plant protein, animal protein, and single-cell protein [8,10].

Because of the imbalanced AA profiles and micronutrients, high indigestible carbohydrate contents, and presence of anti-nutritional factors in plant sources, excess amount of FM replacement with plant protein sources brought about undesirable impacts on the growth and health of fish [7,11]. Compared to plant proteins, thus, animal protein, including terrestrial and fishery by-products, could be a more proper replacer to FM in carnivorous fish feeds because of their substantial quantities of protein and lipid, excellent palatability, and favorable essential AA (EAA) profiles [8,12]. Due to their low price, stable supply, comparatively high protein and lipid content, and qualitatively similar AA profiles to FM, by-products from the processing plants of terrestrial animals, such as livestock and poultry, could be employed as the alternative proteins to FM [13]. However, due to safety issues, such as mad cow disease and swine fever, the use of terrestrial animal by-products in animal feeds has raised public concerns about feed safety [13,14]. 

On the other hand, feed nutritionists have revealed the nutritional values and functional properties of fishery by-products and found that fishery by-products were excellent sources of minerals (Ca, Na, K, P, and Fe), gelatin, and collagen, and its derivatives [15]. Hwang et al. [13] proved that the incorporation of 20% olive flounder skin in the rockfish diets brought about improved growth performance and feed consumption and did not cause any adverse effect on the hematological parameters of rockfish, except for plasma total protein. Furthermore, replacing FM up to 20% with fish waste meal in the diets of African mud catfish (*Clarias gariepinus*) did not deteriorate growth performance, feed consumption, and feed utilization [16].

Tuna has become one of the most popular fish species in recent years because of its outstanding nutritional quality [17]. In 2020, the global catch of the main tuna species was 4.9 million metric tons [18]. Thailand, the Philippines, Ecuador, and Spain are major producers of canned tuna [19]. However, in canning processing plants of tuna, more than half of the total tuna is trashed as waste, including heads, bones, skin, viscera, and fins [20]. Tuna by-product meal (TBM) was recently used as an FM replacer in several fish species, such as olive flounder [21,22,23], rockfish [24,25], and Nile tilapia (*Oreochromis niloticus*) [26]. According to Uyan et al. [23], TBM is a desirable feed ingredient for reducing phosphorus released into the environment, and 50% FM could be substitutable with TBM without retarding the growth of olive flounder. Our previous studies [22,24] demonstrated that up to 40% and 30% FM could be substituted by the fermented blend of TBM and soybean meal (SBM) without deteriorating the growth of rockfish and olive flounder, respectively. Furthermore, Kim et al. [21] recently found that 50% FM could be substituted with TBM in the 65% FM-basal feed of olive flounder without deteriorating the growth and feed availability. Kim et al. [25] recommended dietary 75% FM substitution with TBM in the 64.8% FM-based diet of rockfish when the late stage of juvenile (initial weight of 29.5 g) rockfish were provided with a 64.8% FM-basal feeds or one of the feeds to replace 25%, 50%, 75%, and 100% FM by TBM for 12 weeks. However, fish size (age) can profoundly affect dietary nutrition requirements [27] as well as the substitutability of a replacer for FM in fish diets [28]. In particular, FM up to 10% and 40% in the feeds of juvenile and grower rockfish, respectively, could be substitutable with fermented SBM without deteriorating growth in the 56-day feeding trial [28].

This study, thus, was designed to assess the dietary substitution impacts of FM with TBM on growth, feed availability, hematological chemistry, and biochemical composition of the early stage of juvenile rockfish.

## 2. Materials and Methods

### 2.1. Experimental Design and Diet Preparation

FM was bought from Daekyung Oil & Transportation Co., Ltd. (Busan City, Republic of Korea) (USD 2.11/kg FM, USD = 1304 Won). TBM was bought from Woojinfeed Ind. Co., Ltd. (Incheon City, Republic of Korea) (USD 1.23/kg TBM), which was used as the substitute for FM. 

Six experimental feeds were formulated to be isonitrogenous (51.0%) and isolipidic (12.5%) by adjusting wheat flour and fish oil (Table 1). Specifically, 55% FM and 17.5% fermented SBM were included as the major protein sources in the control (Con) diet to ensure enough protein content for the appropriate growth of rockfish [3,4,25]. Additionally, 19% of wheat flour and 3% of each of fish and soybean oils were contained in the Con diet as the carbohydrate and lipid sources, respectively. Twenty percent increments (20%, 40%, 60%, 80%, and 100%) of FM were replaced with TBM, named the TBM_20_, TBM_40_, TBM_60_, TBM_80_, and TBM_100_ diets, respectively. Additionally, all experimental feeds comprised the same amount of mineral premix (1%), vitamin premix (1%), and choline (0.5%). All experimental feeds were assigned to triplicate groups of rockfish with a complexly randomized design. 

All ingredients were put in a blender and well-blended with water at a ratio of 3:1. The well-mixed ingredients were extruded into pellets using a laboratory pellet extruder (Dongsung Mechanics, Busan City, Republic of Korea) fitted with a 2 mm die hole plate. Then, all diets were dried in an electronic dryer (JW-1350ED; Jinwoo Electronics Co., Ltd., Hwaseong-Si, Gyeonggi-Do, Republic of Korea) at 40 °C for 24 h. The diets were stored in a freezer at −20 °C until use.

### 2.2. Experimental Fish and Feeding Conditions

The early-stage juvenile rockfish were bought from a hatchery (Buan-Gun, Jeollabuk-Do, Republic of Korea). Before the feeding trial, all rockfish were acclimated to the experimental conditions for 10 days by being supplied with commercial pellet (55.0% crude protein and 8.0% crude lipid; National Federation of Fisheries Cooperatives Feeds, Uiryeong-Gun, Gyeongsangnam-Do, Republic of Korea). Upon the completion of the 10-day acclimatization period, 540 juvenile (initial weight of 2.4 g) rockfish were randomly distributed into 18 50 L plastic flow-through tanks, with 30 fish per tank. The proper aeration and 4.3 L/min of the mixture of sand-filtered seawater with underground seawater at a ratio of 1:1 were continuously provided to each tank throughout the feeding experiment. The photoperiod was left under natural circumstances. Rockfish were fed to apparent satiation twice a day (08:00 and 17:00) for 56 days.

### 2.3. Water Quality Monitoring

During the 56-day feeding experiment, water temperature, salinity, dissolved oxygen (DO), and pH were checked every day by a multiparameter water quality meter (AZ-8603, AZ Instrument, Taiwan, China). Water temperature varied from 16.1 to 21.8 °C (19.5 ± 1.67 °C; mean ± SD); salinity varied from 30.8 to 32.4 g/L (31.4 ± 0.35 g/L); DO varied from 7.3 to 8.2 mg/L (7.6 ± 0.24 mg/L), and pH varied from 7.4 to 7.8 (7.5 ± 0.10). The tank bottom was siphon-cleaned every morning, and dead rockfish were immediately eliminated upon observation.

### 2.4. Biological Measurements of Fish

Upon the completion of the 56-day feeding experiment, all live rockfish from each tank were unfed for 24 h and then anesthetized with tricaine methanesulfonate (MS222, 100 mg/L). The total number and final weight of rockfish in each tank were recorded to measure the weight gain, SGR, and survival of rockfish. Eight anesthetized rockfish were individually measured for their total length and weight to determine the condition factor (CF). After that, these eight fish were dissected to weigh the visceral and liver organs to evaluate the viscerosomatic index (VSI) and hepatosomatic index (HSI), respectively. Rockfish biometric parameters were calculated using the following formulas: SGR (%/day) = [(Ln final weight of rockfish − Ln initial weight of rockfish) × 100]/days of feeding trial (56 days); feed efficiency (FE) = (final weight of total rockfish + total weight of dead rockfish − the initial weight of total rockfish)/feed supplied; protein efficiency ratio (PER) = weight gain of rockfish/protein supplied; protein retention (PR, %) = protein gain of rockfish × 100/protein supplied; condition factor (CF, g/cm^3^) = body weight of rockfish (g) × 100/total length of rockfish (cm)^3^; HSI (%) = liver weight of rockfish × 100/body weight of rockfish; VSI (%) = visceral weight of rockfish × 100/body weight of rockfish.

### 2.5. Biochemical Composition of the Experimental Diets and Rockfish

Ten rockfish before the feeding trial and seven rockfish from each tank after the completion of the 56-day feeding trial were homogenized for the whole body’s biochemical composition. The chemical composition and AA and FA profiles of feeds and rockfish were measured using the same procedures and methods previously reported in our study [3]. The AA and FA profiles of main ingredients (FM and TBM) and experimental diets were presented in Table 2 and Table 3, respectively.

### 2.6. Hematological Analysis of Rockfish

Blood samples were drawn via 5 heparinized syringes and 5 syringes from the caudal vein of 10 anesthetized rockfish per tank for the analysis of plasma and serum chemistry of fish, respectively. Plasma and serum were obtained after centrifugation (2700× *g*, 10 min) at 4 °C and then kept in a freezer at −70 °C until analysis. Plasma was used for analysis of aspartate aminotransferase (AST), alanine aminotransferase (ALT), alkaline phosphatase (ALP), total bilirubin (TB), total cholesterol (T-CHO), triglyceride (TG), total protein (TP), and albumin (ALB) with an automatic analyzer (Fuji Dri-Chem NX500i, Fujifilm, Tokyo, Japan). Serum was used for analysis of superoxide dismutase (SOD) and lysozyme activity. The SOD was determined by the ELISA kit (MyBioSource, cat. No. MBS705758, San Diego, CA, USA) according to the manufacturer’s instructions. Absorbance was detected by a microplate reader (Tecan Infinite^®^ 200 PRO, Zürich, Switzerland), and the concentration was calculated by plotting a standard curve. The turbidimetric assay for lysozyme activity was conducted according to Lange et al. [31]. The same procedures and methods were used for lysozyme analysis [3]. 

### 2.7. Economic Analysis of this Study

The economic analysis of this study was performed by using the following formulas, as described by Martínez-Llorens et al. [32]: economic conversion ratio (ECR, USD/kg fish) = feed consumption (kg) × diet price (USD/kg)/weight gain of fish (kg/fish); economic profit index (EPI, USD/fish) = final weight (kg/fish) × fish sale price (USD/kg) − ECR × weight gain (kg/fish). The price (USD/kg) of each ingredient was as follows: FM = 2.11; TBM = 1.23; fermented SBM = 0.66; wheat flower = 0.52; fish oil = 2.61; soybean oil = 1.69; vitamin mix = 7.82; mineral mix = 6.29; and choline = 1.23. The rockfish sale price was calculated at USD 9.53/kg [1].

### 2.8. Statistical Analysis

One-way ANOVA and Tukey’s multiple comparison tests were used to identify significant differences (*p* < 0.05) among dietary treatments in SPSS version 26.0 (SPSS Inc., Chicago, IL, USA). Before statistical analysis, all percentage data were arcsine-transformed. The orthogonal polynomial contrasts were conducted to determine the response for all dependent variables (linear, quadratic, or cubic), and regression analysis was conducted to find the best-fitting model when a significant difference was found.

## 3. Results

### 3.1. AA and FA Profiles of the Main Ingredients and Experimental Feeds

All essential AAs (EAA), except for histidine and tryptophan, and non-EAAs (NEAA), except for glycine and proline in FM, were relatively higher than those in TBM. Histidine and tryptophan content tended to increase with dietary increased FM replacement levels with TBM but decrease for the rest of all EAA in all experimental diets. The requirement of lysine (2.99% of the diet) [29] for the growth of juvenile rockfish was met in all experimental diets, while the requirement of methionine for rockfish (1.37% of the diet) [4] was not met in all experimental diets (1.05–1.13% of the diet). Leucine and glutamic acid were the highest EAA and NEAA in all experimental feeds, respectively.

The total content of saturated FA (∑SFA) and monounsaturated FA (∑MUFA) in FM were relatively low over those in TBM but high for total content of n–3 highly unsaturated FA (∑n-3 HUFA), including eicosapentaenoic acid (EPA, 20:5n-3), and docosahexanoic acid (DHA, 22:6n-3) content. With increased FM substitution levels with TBM, the ∑SFA and ∑MUFA in all experimental feeds were elevated but lowered for the ∑n-3 HUFA. Nevertheless, the dietary ∑n-3 HUFA requirement of rockfish (7.20% of total FA) [30] was fulfilled in all experimental feeds (9.50%–11.92% of total FA). 

### 3.2. Performance of Rockfish

The survival of rockfish (≥95.6%) was not considerably (*p >* 0.8) impacted by dietary treatments (Table 4). However, rockfish fed the Con, TBM_20_, and TBM_40_ diets achieved superior (*p* < 0.001) weight gain to rockfish fed all other diets. Rockfish fed the TBM_20_ diet achieved superior (*p* < 0.001) SGR to rockfish fed the TBM_60_, TBM_80_, and TBM_100_ diets but comparable to rockfish fed the Con and TBM_40_ diets. Furthermore, the SGR of rockfish fed the Con diet was also comparable to that of fish fed the TBM40 and TBM60 diets. In terms of the orthogonal contrast, notable linear (*p* = 0.001 and *p* = 0.002, respectively) and quadratic (*p* = 0.021 and *p* = 0.037, respectively) relationships were exhibited in weight gain and SGR of rockfish versus dietary FM replacement levels with TBM. The most suitable models between dietary FM replacement levels with TBM and weight gain (Y = −0.000272X^2^ + 0.001185X + 10.2893; adjusted R^2^ = 0.8729; *p* < 0.001) and SGR (Y = −0.000045X^2^ + 0.000657X + 2.9936; adjusted R^2^ = 0.8448; *p* < 0.001) of rockfish were observed (Figure 1). 

### 3.3. Feed Availability and Biometric Indices of Rockfish

Feed intake of rockfish fed the Con, TBM_20_, and TBM_40_ diets were superior (*p* < 0.001) to rockfish fed all other diets (Table 5). In regard to the orthogonal contrast, a significant linear (*p* = 0.001) relationship was presented in feed intake of rockfish versus dietary FM replacement levels with TBM. The most suitable model between dietary FM replacement levels with TBM and feed intake of rockfish (linear, *p* < 0.001, R^2^ = 0.752) was observed.

Values (means of triplicate) in the same row sharing the same superscript letter are not significantly different (*p >* 0.05). Abbreviations: SEM, pooled standard error of treatment means; Adj. R^2^, adjusted R square; L, linear; Q, quadratic; NR, no relationship. ^1^ Feed efficiency (FE) = (final weight of total rockfish + total weight of dead rockfish—initial weight of total rockfish)/feed supplied. ^2^ Protein efficiency ratio (PER) = Weight gain of rockfish/protein supplied. ^3^ Protein retention (PR, %) = Protein gain of rockfish ×100/protein supplied. ^4^ Condition factor (CF, g/cm^3^) = Body weight of rockfish (g) × 100/total length of rockfish (cm)^3^. ^5^ Hepatosomatic index (HSI, %) = Liver weight of rockfish × 100/body weight of rockfish. ^6^ Visceralsomatic index (VSI, %) = Viscera weight of rockfish × 100/body weight of rockfish.

FE and PR of rockfish were not considerably altered (*p* > 0.2 and *p* > 0.1, respectively) by dietary treatments. However, rockfish fed the TBM_20_, TBM_40_, and TBM_60_ diets achieved considerably (*p* < 0.005) higher PER than rockfish fed the TBM_100_ diet but not considerably (*p* > 0.05) different from that of rockfish fed the Con and TBM_80_ diets. In regard to the orthogonal polynomial contrast, significant linear (*p* = 0.001) and quadratic (*p* = 0.007) relationships were shown in PER of rockfish versus dietary FM replacement levels with TBM. The most suitable model between dietary FM substitution levels with TBM and PER of rockfish (quadratic, *p* < 0.001, R^2^ = 0.587) was observed.

The biometric indices of CF, HSI, and VSI of rockfish were not considerably altered (*p* > 0.4, *p* > 0.3, and *p* > 0.1, respectively) by dietary FM substitution with TBM.

### 3.4. Biochemical Composition of the Whole-Body Rockfish

The moisture (73.2–73.8%), crude protein (14.9–15.7%), crude lipid (6.3–6.9%), and ash (3.7–4.5%) content of the whole-body rockfish were not considerably (*p* > 0.9, *p* > 0.5, *p* > 0.2, and *p* > 0.5, respectively) impacted by dietary treatments (Table 6).

The whole-body rockfish’s AA profiles were not considerably (*p* > 0.05 for all) impacted by dietary FM substitution by TBM (Table 7).

The ∑MUFA of rockfish tended to increase with dietary increased FM replacement by TBM, except for the TBM_20_ diet, but decrease for the ∑n-3 HUFA and EPA (Table 8). Furthermore, the ∑n-3 HUFA in rockfish fed the Con, TBM_20_, and TBM_40_ diets were considerably (*p* < 0.002) higher than that of fish fed the TBM_100_ diet but not considerably (*p* > 0.05) different from that of rockfish fed the TBM_60_ and TBM_80_ diets. In regard to the orthogonal polynomial contrast, notable linear (*p* = 0.001 for both) relationships and linear (*p* = 0.001), quadratic (*p* = 0.001), and cubic (*p* = 0.001) relationships were shown in the ∑SFA, ∑n-3 HUFA, and ∑MUFA of rockfish, respectively. The most suitable models between dietary FM replacement levels with TBM and ∑SFA (linear, *p* < 0.001, R^2^ = 0.534), ∑MUFA (quadratic, *p* < 0.001, R^2^ = 0.951), and ∑n-3 HUFA of rockfish (linear, *p* < 0.001, R^2^ = 0.689) were observed.

### 3.5. Hematological Parameters of Rockfish

All plasma measurements of rockfish, such as AST (148.7–150.7 U/L), ALT (23.0–24.3 U/L), ALP (183.7–185.7 U/L), TB (1.1–1.8 mg/dL), T-CHO (247.0–253.3 mg/dL), TG (360.7–364.0 mg/dL), TP (4.3–4.8 g/dL), and ALB (0.9–1.4 g/dL) were not considerably (*p* > 0.05 for all) changed by dietary FM replacement with TBM (Table 9).

The serum SOD (2.4–2.7 ng/mL) and lysozyme activity (282.5–399.9 U/mL) were not considerably (*p* > 0.6 and *p* > 0.5, respectively) altered by dietary substitution of TBM for FM.

### 3.6. Economic Analysis of this Study

Increased FM replacement with TBM changed the diet price and economic parameters (ECR and EPI) in this study (Table 10). The price of the experimental feeds decreased with increased FM replacement levels with TBM. The ECR of the TBM_100_ diet was considerably (*p* < 0.0001) lower than that of the Con, TBM_20_, and TBM_40_ diets but not considerably (*p* > 0.05) different from that of the TBM_60_ and TBM_80_ diets. However, the EPI of the Con, TBM_20_, and TBM_40_ diets were considerably (*p* < 0.0001) higher than that of the TBM_60_, TBM_80_, and TBM_100_ diets. In regard to orthogonal contrast, notable linear (*p* = 0.001 for both) and quadratic (*p* = 0.035 and *p* = 0.003, respectively) relationships were shown in ECR and EPI of diets versus dietary replacement of TBM for FM. The most suitable models between dietary FM replacement level with TBM and ECR (linear, *p* < 0.001, R^2^ = 0.878) and EPI of diets (quadratic, *p* < 0.001, R^2^ = 0.839) were observed.

## 4. Discussion

No remarkable differences in weight gain and feed consumption of rockfish fed the Con and TBM_40_ diets in the present study implied that FM up to 40% could be replaceable with TBM in the diets of the early stage of juvenile rockfish grown from 2.4 g to ca. 10 g. In addition, no discernible difference in the SGR of rockfish fed the Con and TBM60 diets also indicated that FM up to 60% could be replaceable with TBM without deteriorating the SGR of rockfish. Unlike this study, however, Kim et al. [25] demonstrated that FM up to 75% could be replaceable with TBM in the diets of rockfish without deterioration of growth when the late-stage juvenile rockfish grown from 29 g to 53 g were provided with a 64.8% FM-basal feed or one of the feeds substituting 25, 50, 75, and 100% FM by TBM for 12 weeks. This disparity could be a result of the fish size (initial weight of 2.4 g in this study vs. initial weight of 29 g in Kim et al.’s study [25]). The smaller fish seemed to be highly limited to lower amounts of FM substitution with alternative feeds than the larger ones [28,33]. FM replacement up to 10% and 40% could be successfully made in the diets of juvenile (initial weight of 1.2 g) and grower (initial weight of 148.2 g) rockfish, respectively, without deteriorating growth, feed intake, and PER when fish were fed with a 58% FM-based diet or diets substituting 10, 20, 30, and 40% FM with fermented SBM for 56 days [28]. In addition, Burr et al. [33] suggested that the substitution of FM with alternative protein blends in diets for the early-stage juvenile (5.5 g) salmon was not recommendable, but FM could be replaced with land animal protein ingredients in diets for the late stage juvenile (over 30 g) salmon. Relatively higher SGR values (2.63–3.01%/day) of rockfish were obtained in this study compared to those (0.60–0.69%/day) of rockfish in Kim et al.’s study [25]. The relatively low growth rate of rockfish in Kim et al.’s study [25] could be another reason why the substitutability of TBM for FM in rockfish feeds in Kim et al.’s study [25] was higher (75% FM substitution) than in this study (40% FM substitution).

Weight gain and feed consumption of rockfish fed diets replacing higher than 40% with TBM were remarkably reduced in this experiment, being in accordance with the findings of the previous research [23,34,35] and showing that feed intake of fish had an inverse relationship to dietary optimum FM substitution levels with the alternative proteins. Deteriorated palatability of diets usually results in decreased feed consumption, which eventually leads to reduced growth in fish [28,34,35]. Alanine showed a strong attractiveness to rockfish [36]. With increased FM substitution by TBM in diets, alanine content decreased in the present study, and this might be one of the reasons why dietary high (60–100%) FM replacement with TBM led to reduced feed intake of rockfish. Furthermore, the feed intake of olive flounder could also be influenced by dietary lipid content and FA profiles [23]. Thus, the inclusion of palatability enhancers in low FM diets may be a good feeding strategy to improve the feed intake of fish.

The AA profiles of diets are very important in formulating low FM diets, as deficiencies of EAA may lead to poor growth of fish [28,34,37]. A deficiency of one or more EAA in diets typically limits the adsorption of other AAs and may cause these AAs to be utilized as energy sources [38]. For instance, insufficient lysine content in diets led to low feed utilization and poor growth of seabass (*Lateolabrax japonicus*) [39], rainbow trout (*Oncorhynchus mykiss*) [40], and grass carp (*Ctenopharyngodon idellus*) [41]. However, the EAA requirements could also be altered by several factors, such as feed composition, water temperature, fish size (age), and sex [37]. Lysine content (3.01–3.13% of the diet) in all experimental feeds in the present experiment satisfied the dietary lysine requirement (2.99% of the diet) of rockfish [29]. Several studies [42,43] have already demonstrated that cysteine could spare about 40–50% of the dietary methionine requirements of fish. Although the content of methionine (1.05–1.13% of the diet) in all experimental feeds was relatively low over the methionine requirement (1.37% of the diet) for rockfish [4], the cysteine content (0.65–0.69% of the diet) of all experimental feeds would be much higher than its content (0.12% of the diet) reported by Yan et al. [4]. Thus, relatively low methionine content in all experimental feeds in this study might not have a negative impact on the growth of rockfish. Additionally, marine finfish require an adequate amount of n-3 HUFA to achieve appropriate growth [44]. Although the ∑n-3 HUFA in TBM was relatively low over that in FM, the ∑n-3 HUFA content (9.50–11.52% of total FA) in all experimental feeds would meet dietary ∑n-3 HUFA requirement (7.20% of total FA) of rockfish [30].

Fishery by-products have different nutritional qualities than FM produced from whole fish, typically lower in protein but higher in minerals [45]. However, approximately 52–54% of the total tuna was generated as by-products and discarded as waste after the canning process [20]. Our previous study [21] unveiled that 50% FM could be substituted by TBM in the 65% FM-basal feed of olive flounder without deteriorating growth and feed utilization, and estimated dietary optimum FM substitution levels with TBM of 47.1%, 48.6%, and 46.3% according to weight gain, SGR, and FE, respectively. Likewise, Hernández et al. [46] found that replacing up to 40% FM with TBM in the juvenile spotted rose snapper (*Lutjanus guttatus*) feeds could be possible without negatively influencing the growth and feed utilization. Furthermore, Uyan et al. [23] reported that up to 50% FM replacement by tuna muscle by-product powder (TMP) in diets did not compromise the growth and feed utilization of olive flounder, and compared to fish fed the 58.5% FM-basal diet, dietary 50% FM replacement by TMP led to 50% lower *p* discharged into the surroundings. Tuna by-products seem to be an appropriate replacer for FM in several fish diets. 

The PER of rockfish fed the TBM_20_, TBM_40_, and TBM_60_ feeds were greater than that of rockfish fed the TBM_100_ diet but comparable to rockfish fed the Con and TBM_80_ diets, whereas the FE and PR of rockfish were not influenced by dietary FM replacement by TBM. This might indicate that FM up to 80% could be substituted by TBM without negatively influencing feed utilization (FE, PER, and PR) of rockfish. Similarly, in the 12-week feeding experiment, the FE and PER of rockfish were not influenced when late-stage juvenile (initial weight of 29.5) rockfish were fed with a 64.8% FM-basal diet or one of diets substituting 25, 50, and 75 FM by TBM [25], but dietary 100% FM substitution with TBM lowered FE and PER. Moreover, feed utilization (feed conversion ratio and PER) of spotted rose snapper was not altered by dietary FM substitution with TBM [46].

In considering the economic analysis of this study, the price of the experimental diets was lowered with increased FM substitution levels with TBM because of the lower price of TBM compared to FM, which eventually led to a reduced ECR. However, the higher EPI of the Con, TBM_20_, and TBM_40_ feeds compared to that of TBM_60_, TBM_80_, and TBM_100_ diets also indicated that the former could produce greater economic returns than the latter to farmers. The TBM_40_ diet with relatively low ECR was the most recommendable to farmers. This result also supported the results of weight gain and feed intake of rockfish based on multiple comparisons in this study.

The biometric indices can reveal information on the physiological and nutritional status of fish [47]. However, no differences in CF, HSI, and VSI of rockfish were found among dietary treatments. Likewise, dietary FM substitution with squid (*Sepia esculenta*) liver powder [48] and meat meal [3] did not cause any differences in the biometric indices of rockfish. However, there were also some conflicting results [21,47,49] demonstrating that biometric indices of fish were impacted by dietary FM substitution with poultry by-product meal, TBM, and fermented plant protein sources. 

The whole-body proximate composition and AA profiles of rockfish did not exhibit considerable variability with dietary FM replacement with TBM, being consistent with previous findings that dietary FM replacement with TBM [21,25] or the combined TBM with plasma powder [50] did not change the fish whole-body’s proximate composition and/or AA profiles. Ai et al. [51] demonstrated that fish could maintain steady protein content and EAA profiles in their bodies to maintain normal physiological function when they were subjected to different diets or environmental challenges. However, the dry matter and crude lipid of olive flounder [23] and crude lipid of rockfish [25] were altered by dietary FM replacement with TBM. 

The EPA and DHA belong to the n-3 HUFA family, and they are non-dispensable nutrients for all vertebrates, including fish and humans [52]. Seafood or other aquaculture products are the primary sources of EPA and DHA, with great benefits for human health [53]. Although the content of ∑n-3 HUFA met the requirement for rockfish in all experimental diets, the FA profiles of the whole body of fish were changed by dietary FM replacement by TBM in this study. Due to the relatively low content of ∑n-3 HUFA, including EPA, but similar content of DHA in TBM over those in FM, the ∑n-3 HUFA, including EPA in the whole-body rockfish tended to decrease with dietary increased FM replacement by TBM, whereas DHA content was not altered by dietary FM replacement by TBM. The ∑MUFA of the whole body of rockfish, except for rockfish fed the TBM_20_ diet, and the ∑n-3 HUFA were well reflected from the FA profiles of the experimental feeds, agreeing with previous studies [52,54], showing that the fish whole body’s FA profiles were closely related to dietary FA profiles. Likewise, high or complete replacement of FM with alternative proteins influenced the FA profiles of whole-body rainbow trout (*Oncorhynchus mykiss*) [55] and Jian carp (*Cyprinus carpio*) [56]. However, Lee et al. [3] revealed that partial or complete substitution of FM with meat meal in feeds did not change the FA profiles of the whole body of rockfish.

Plasma parameters are trustworthy gauges of the physiological condition and welfare of fish [14]. Among all plasma parameters, AST, ALT, and ALP are marker enzymes to evaluate the condition of the liver [57,58]. In general, high AST and ALT activities suggest a deterioration or impairment of normal liver function [59]. Habte-Tsion et al. [59] found that AST and ALT of blunt snout bream (*Megalobrama amblyephala*) were negatively affected by the dietary excess in threonine levels. However, none of the plasma parameters were changed by dietary FM replacement with TBM in the present experiment, agreeing with previous studies demonstrating that the plasma chemistry of olive flounder was unaffected by the dietary FM substitution by TBM [21] or combined plasma powder and TBM [50]. This result might imply that dietary FM replacement with TBM had no detrimental effect on the plasma condition of rockfish. Likewise, the dietary FM replacement with meat meal [3], poultry by-product meal [47], and combined plasma powder and chicken by-product meal [50] did not impact plasma measurements of rockfish, black sea bream (*Acanthoparus schlegelii*), and olive flounder, respectively. However, unlike this study, plasma T-CHO of olive flounder was changed by dietary FM substitution by cricket (*Gryllus bimaculatus*) meal and TBM, respectively [22,60].

SOD is an enzyme that acts as an antioxidant, preventing free radicals from damaging animal tissue through neutralization [50]. Lysozyme is a crucial defense molecule because of its mediating function against microbial invasion [61]. SOD and lysozyme activity are trustworthy evidence for evaluating the impact of nutritional treatments on the health status of fish [49]. The serum SOD and lysozyme activity of rockfish were not altered by dietary FM replacement by TBM in this experiment. This might imply that dietary FM replacement with TBM did not cause any detrimental influences on SOD and lysozyme activity of rockfish. Likewise, dietary FM substitution with alternative proteins did not affect the SOD and lysozyme activity of olive flounder [3,21] and lysozyme activity of Atlantic salmon [62] and European seabass (*Dicentrarchus labrax*) [54]. However, SOD and lysozyme activity of yellow catfish and hybrid tilapia (*Oreochromis niloticus* × *O. aureus*) were altered with dietary FM substitution with black soldier fly (*Hermetia illucens*) larvae meal [63] and SBM [64], respectively. No differences in plasma and serum chemistry of rockfish might imply that dietary FM replacement with TBM had no detrimental impact on hematological measurements in the present experiment. Likewise, partial substitution of FM with alternative sources in diets did not exhibit any negative impact on the health status and/or meat quality of fish [65,66,67].

## 5. Conclusions

FM up to 40% could be substitutable with TBM in the diets of rockfish based on weight gain and feed consumption. The TBM_40_ diet produced the highest EPI with relatively low ECR. Therefore, the TBM_40_ diet seemed to be the most recommendable strategy to farmers. 

## Figures and Tables

**Figure 1 animals-13-03586-f001:**
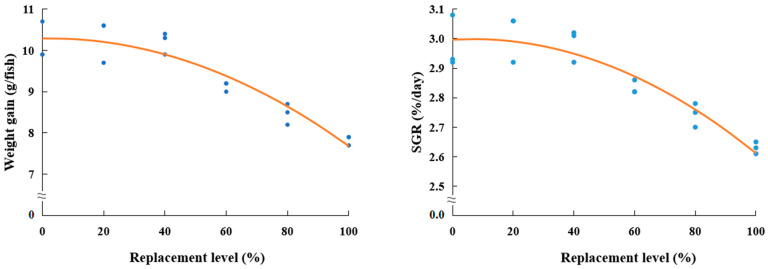
Quadratic relationships between dietary fish meal replacement levels (%) with tuna by-product meal and weight gain (g/fish) (Y = −0.000272X^2^ + 0.001185X + 10.2893, adjusted R^2^ = 0.8729, *p* < 0.001), and specific growth rate (SGR, %/day) (Y = −0.000045X^2^ + 0.000657X + 2.9936, adjusted R^2^ = 0.8448, *p* < 0.001) of rockfish (*Sebastes schlegeli*).

**Table 1 animals-13-03586-t001:** Ingredient and chemical composition of the experimental diets (%, dry matter basis).

	Experimental Diets					
	Con	TBM_20_	TBM_40_	TBM_60_	TBM_80_	TBM_100_
Ingredient (%, DM)						
Fish meal (FM) ^1^	55.0	44.0	33.0	22.0	11.0	0.0
Tuna by-product meal (TBM) ^2^		12.8	25.6	38.4	51.2	64.0
Fermented soybean meal	17.5	17.5	17.5	17.5	17.5	17.5
Wheat flour	19.0	17.1	15.2	13.3	11.4	9.5
Fish oil	3.0	3.1	3.2	3.3	3.4	3.5
Soybean oil	3.0	3.0	3.0	3.0	3.0	3.0
Mineral premix ^3^	1.0	1.0	1.0	1.0	1.0	1.0
Vitamin premix ^4^	1.0	1.0	1.0	1.0	1.0	1.0
Choline	0.5	0.5	0.5	0.5	0.5	0.5
Nutrients (%, DM)						
Dry matter	98.4	98.2	98.0	98.6	98.2	98.4
Crude protein	50.9	50.4	50.5	50.5	51.2	51.3
Crude lipid	12.7	12.1	12.2	12.9	12.4	12.7
Ash	11.4	12.1	13.2	12.9	13.7	14.6

^1^ Fish meal (FM) (crude protein: 69.7%; crude lipid: 11.1%; ash: 16.7%) imported from Peru was purchased from Daekyung Oil & Transportation Co., Ltd. (Busan City, Republic of Korea) [USD 2.11/kg FM, USD = 1304 Won (Korean currency)]. ^2^ Tuna by-product meal (TBM) (crude protein: 62.3%; crude lipid: 9.5%; ash: 18.6%) was purchased from Woojinfeed Ind. Co., Ltd. (Incheon City, Republic of Korea) (USD 1.23/kg TBM). ^3^ Mineral premix contained the following ingredients (g/kg mix): MgSO_4_·7H_2_O, 80.0; NaH_2_PO_4_·2H_2_O, 370.0; KCl, 130.0; ferric citrate, 40.0; ZnSO_4_·7H_2_O, 20.0; Ca-lactate, 356.5; CuCl, 0.2; AlCl_3_·6H_2_O, 0.15; KI, 0.15; Na_2_Se_2_O_3_, 0.01; MnSO_4_·H_2_O, 2.0; CoCl_2_·6H_2_O, 1.0. ^4^ Vitamin premix contained the following amount which were diluted in cellulose (g/kg mix): L-ascorbic acid, 121.2; DL-α-tocopheryl acetate, 18.8; thiamin hydrochloride, 2.7; riboflavin, 9.1; pyridoxine hydrochloride, 1.8; niacin, 36.4; Ca-D-pantothenate, 12.7; myo-inositol, 181.8; D-biotin, 0.27; folic acid, 0.68; p-aminobenzoic acid, 18.2; menadione, 1.8; retinyl acetate, 0.73; cholecalciferol, 0.003; cyanocobalamin, 0.003.

**Table 2 animals-13-03586-t002:** Amino acid (AA) profiles (% of the diet) of the main ingredients (FM and TBM) and experimental diets replacing various levels of fish meal with tuna by-product meal.

	Ingredients		Requirement	Experimental Diets					
	FM	TBM		Con	TBM_20_	TBM_40_	TBM_60_	TBM_80_	TBM_100_
Essential amino acid (%)									
Arginine	3.51	3.47		2.61	2.60	2.57	2.57	2.55	2.52
Histidine	1.50	1.56		1.03	1.10	1.13	1.17	1.19	1.22
Isoleucine	2.47	2.37		1.70	1.69	1.68	1.68	1.66	1.65
Leucine	4.50	4.12		3.59	3.52	3.47	3.42	3.41	3.34
Lysine	4.82	4.01	2.99 ^1^	3.13	3.11	3.11	3.09	3.06	3.01
Methionine	1.75	1.28	1.37 ^2^	1.13	1.07	1.07	1.06	1.05	1.05
Phenylalanine	2.42	2.30		1.97	1.96	1.95	1.93	1.93	1.90
Threonine	2.69	2.53		2.00	1.99	1.95	1.94	1.92	1.91
Tryptophan	0.49	0.63		0.33	0.38	0.41	0.41	0.42	0.50
Valine	2.96	2.85		2.01	2.01	2.00	1.99	1.97	1.96
^3^ ∑EAA	27.11	25.12		19.50	19.43	19.34	19.26	19.16	19.06
Non-essential amino acid (%)									
Alanine	4.04	3.98		2.88	2.87	2.85	2.85	2.83	2.82
Aspartic acid	5.58	5.12		4.33	4.30	4.29	4.26	4.26	4.25
Cysteine	0.80	0.60	0.12 ^2^	0.69	0.68	0.67	0.66	0.66	0.65
Glutamic acid	8.08	7.09		7.15	7.14	7.00	6.82	6.68	6.63
Glycine	3.94	4.51		2.72	2.94	2.95	3.12	3.25	3.35
Proline	2.65	2.95		2.39	2.39	2.40	2.43	2.44	2.44
Serine	2.46	2.39		2.08	2.06	2.06	2.03	2.01	2.00
Tyrosine	1.53	1.48		1.19	1.16	1.16	1.15	1.12	1.11
^4^ ∑NEAA	29.08	28.12		23.43	23.54	23.38	23.32	23.25	23.25

^1^ Data were obtained from Yan et al.’s study [29]. ^2^ Data were obtained from Yan et al.’s study [4]. ^3^ ∑EAA: total content of essential amino acids. ^4^ ∑NEAA: total content of non-essential amino acids.

**Table 3 animals-13-03586-t003:** Fatty acids (FA) profiles (% of total fatty acids) of the main ingredients (FM and TBM) and experimental diets replacing various levels of fish meal with tuna by-product meal.

	Ingredients		Requirement	Experimental Diets					
	FM	TBM		Con	TBM_20_	TBM_40_	TBM_60_	TBM_80_	TBM_100_
C14:0	5.93	4.58		2.90	2.90	2.75	2.63	2.55	2.42
C16:0	23.07	25.59		17.00	17.39	17.58	17.71	17.85	18.11
C18:0	4.18	5.96		3.81	4.04	4.20	4.25	4.44	4.68
C20:0	0.13	0.07		0.19	0.19	0.19	0.18	0.16	0.17
C22:0	0.19	0.33		0.28	0.29	0.30	0.32	0.33	0.34
C24:0	0.74	0.74		0.61	0.59	0.59	0.60	0.63	0.66
∑SFA ^1^	34.24	37.27		24.79	25.40	25.61	25.69	25.96	26.38
C16:1n-7	6.82	5.25		3.83	3.81	3.80	3.72	3.68	3.58
C18:1n-9	14.38	21.73		25.66	25.96	26.64	27.68	27.92	28.77
C20:1n-9	2.05	2.96		1.54	1.66	1.73	1.80	1.86	1.96
C22:1n-9	0.12	0.10		0.03	0.03	0.02	0.02	0.02	0.02
C24:1n-9	1.61	1.23		0.78	0.78	0.76	0.74	0.73	0.70
∑MUFA ^2^	24.98	31.27		31.84	32.24	32.95	33.96	34.21	35.03
C18:2n-6	2.95	1.82		23.55	22.40	21.93	21.50	21.15	20.70
C18:3n-3	0.87	0.68		3.21	3.15	3.10	3.09	3.07	2.93
C20:4n-6	3.90	4.67		2.19	2.27	2.30	2.30	2.31	2.35
C20:5n-3	12.76	5.16		4.96	4.68	4.29	3.79	3.40	2.85
C22:2n-6	0.65	0.46		0.34	0.32	0.30	0.28	0.27	0.25
C22:6n-3	15.84	13.93		6.96	6.93	6.92	6.82	6.82	6.65
∑n-3 HUFA ^3^	28.60	19.09	7.20 ^4^	11.92	11.61	11.21	10.61	10.22	9.50
Unknown	3.81	4.74		2.16	2.61	2.60	2.57	2.81	2.86

^1^ ∑SFA: total content of saturated fatty acids. ^2^ ∑MUFA: total content of monounsaturated fatty acids. ^3^ ∑n-3 HUFA: total content of n-3 highly unsaturated fatty acids. ^4^ Data were obtained from Lee et al.’s study [30].

**Table 4 animals-13-03586-t004:** Survival (%), weight gain (g/fish), and specific growth rate (SGR) of rockfish fed the experimental diets replacing various levels of fish meal with tuna by-product meal for 56 days.

	Experimental Diets								Orthogonal Contrast			Regression		
	Con	TBM_20_	TBM_40_	TBM_60_	TBM_80_	TBM_100_	SEM	*p*-Value	Linear	Quadratic	Cubic	Model	*p*-Value	Adj. R^2^
Initial weight (g/fish)	2.4	2.3	2.4	2.4	2.3	2.3	0.01	-	-	-	-	-	-	-
Final weight (g/fish)	12.6 ^a^	12.6 ^a^	12.5 ^a^	11.5 ^b^	10.8 ^bc^	10.2 ^c^	0.24	<0.001	0.001	0.026	0.171	Q	<0.001	0.894
Survival (%)	95.6	95.6	95.6	96.7	97.8	96.7	0.66	>0.8	0.401	0.950	0.564	NR	-	-
Weight gain (g/fish)	10.2 ^a^	10.3 ^a^	10.2 ^a^	9.1 ^b^	8.5 ^bc^	7.8 ^c^	0.24	<0.001	0.001	0.021	0.108	Q	<0.001	0.873
SGR ^1^ (%/day)	2.97 ^ab^	3.01 ^a^	2.98 ^ab^	2.83 ^bc^	2.74 ^cd^	2.63 ^d^	0.04	<0.001	0.002	0.037	0.134	Q	<0.001	0.845

Values (means of triplicate) in the same row sharing the same superscript letter are not significantly different (*p >* 0.05). Abbreviations: SEM, pooled standard error of treatment means; Adj. R^2^, adjusted R square; Q, quadratic; NR, no relationship. ^1^ Specific growth rate (SGR, %/day) = [(Ln final weight of rockfish − Ln initial weight of rockfish) ×100]/days of feeding trial.

**Table 5 animals-13-03586-t005:** Feed consumption (g/fish), feed efficiency (FE), protein efficiency ratio (PER), protein retention (PR), condition factor (CF), hepatosomatic index (HSI), and viscerosomatic index (VSI) of rockfish fed the experimental diets replacing various levels of fish meal with tuna by-product meal for 56 days.

	Experimental Diets								Orthogonal Contrast			Regression		
	Con	TBM_20_	TBM_40_	TBM_60_	TBM_80_	TBM_100_	SEM	*p*-Value	Linear	Quadratic	Cubic	Model	*p*-Value	Adj. R^2^
Feed consumption (g/fish)	9.70 ^a^	9.65 ^a^	9.61 ^a^	8.58 ^b^	8.47 ^b^	7.92 ^b^	0.18	< 0.001	0.001	0.153	0.211	L	<0.001	0.752
FE ^1^	1.04	1.05	1.05	1.04	1.00	1.00	0.01	>0.2	0.038	0.266	0.381	NR	-	-
PER ^2^	2.06 ^ab^	2.11 ^a^	2.10 ^a^	2.11 ^a^	1.96 ^ab^	1.93 ^b^	0.02	<0.005	0.001	0.007	0.452	Q	<0.001	0.587
PR (%) ^3^	32.3	31.7	33.2	31.5	30.0	28.5	0.55	>0.1	0.021	0.145	0.970	NR	-	-
CF (g/cm^3^) ^4^	1.55	1.49	1.52	1.56	1.60	1.61	0.02	>0.4	0.124	0.400	0.318	NR	-	-
HSI (%) ^5^	2.18	2.39	2.63	2.25	2.58	2.59	0.07	>0.3	0.138	0.673	0.320	NR	-	-
VSI (%) ^6^	9.44	9.41	9.88	9.68	10.00	9.94	0.08	>0.1	0.015	0.649	0.714	NR	-	-

Values (means of triplicate) in the same row sharing the same superscript letter are not significantly different (*p >* 0.05). Abbreviations: SEM, pooled standard error of treatment means; Adj. R^2^, adjusted R square; L, linear; Q, quadratic; NR, no relationship. ^1^ Feed efficiency (FE) = (final weight of total rockfish + total weight of dead rockfish—initial weight of total rockfish)/feed supplied. ^2^ Protein efficiency ratio (PER) = Weight gain of rockfish/protein supplied. ^3^ Protein retention (PR, %) = Protein gain of rockfish ×100/protein supplied. ^4^ Condition factor (CF, g/cm^3^) = Body weight of rockfish (g) × 100/total length of rockfish (cm)^3^. ^5^ Hepatosomatic index (HSI, %) = Liver weight of rockfish × 100/body weight of rockfish. ^6^ Visceralsomatic index (VSI, %) = Viscera weight of rockfish × 100/body weight of rockfish.

**Table 6 animals-13-03586-t006:** Proximate composition (% of wet weight) of rockfish fed the experimental diets replacing various levels of fish meal with tuna by-product meal for 56 days.

	Experimental Diets								Orthogonal Contrast			Regression		
	Con	TBM_20_	TBM_40_	TBM_60_	TBM_80_	TBM_100_	SEM	*p*-Value	Linear	Quadratic	Cubic	Model	*p*-Value	Adj. R^2^
Moisture	73.2	73.3	73.6	73.3	73.3	73.8	0.16	>0.9	0.496	0.896	0.572	NR	-	-
Crude protein	15.6	15.0	15.7	15.0	15.3	14.9	0.16	>0.5	0.264	0.830	0.670	NR	-	-
Crude lipid	6.9	6.5	6.3	6.6	6.5	6.3	0.08	>0.2	0.114	0.451	0.095	NR	-	-
Ash	4.3	3.7	4.4	4.3	4.5	4.4	0.12	>0.5	0.311	0.762	0.304	NR	-	-

Abbreviations: SEM, pooled standard error of treatment means; Adj. R^2^, adjusted R square; NR, no relationship.

**Table 7 animals-13-03586-t007:** AA profiles (% of wet weight) of the whole body of rockfish fed the experimental diets, replacing various levels of fish meal with tuna by-product meal for 56 days.

	Experimental Diets								Orthogonal Contrast			Regression		
	Con	TBM_20_	TBM_40_	TBM_60_	TBM_80_	TBM_100_	SEM	*p*-Value	Linear	Quadratic	Cubic	Model	*p*-Value	Adj. R^2^
Essential amino acid (%)														
Arginine	0.94	0.96	0.90	0.95	0.94	0.90	0.01	>0.5	0.376	0.682	0.452	NR	-	-
Histidine	0.32	0.32	0.31	0.32	0.31	0.31	0.01	>0.9	0.474	0.985	0.990	NR	-	-
Isoleucine	0.58	0.58	0.55	0.57	0.55	0.56	0.01	>0.9	0.589	0.820	0.899	NR	-	-
Leucine	1.08	1.08	1.03	1.07	1.03	1.05	0.02	>0.8	0.473	0.660	0.944	NR	-	-
Lysine	1.20	1.18	1.12	1.20	1.15	1.16	0.01	>0.4	0.429	0.498	0.464	NR	-	-
Phenylalanine	0.57	0.58	0.59	0.58	0.57	0.57	0.01	>0.9	0.878	0.631	0.702	NR	-	-
Threonine	0.70	0.70	0.67	0.69	0.68	0.68	0.01	>0.9	0.708	0.864	0.969	NR	-	-
Tryptophan	0.10	0.08	0.09	0.09	0.10	0.09	0.01	>0.9	0.959	0.770	0.744	NR	-	-
Valine	0.67	0.67	0.64	0.66	0.64	0.64	0.01	>0.9	0.501	0.880	0.976	NR	-	-
Non-essential amino acid (%)														
Alanine	1.03	1.05	1.03	1.06	1.05	0.99	0.02	>0.9	0.713	0.375	0.625	NR	-	-
Aspartic acid	1.47	1.47	1.39	1.46	1.43	1.43	0.02	>0.7	0.467	0.633	0.768	NR	-	-
Glutamic acid	2.03	2.04	1.95	2.02	2.00	1.98	0.02	>0.7	0.511	0.760	0.680	NR	-	-
Glycine	1.20	1.28	1.25	1.26	1.31	1.13	0.02	>0.1	0.549	0.400	0.359	NR	-	-
Proline	0.69	0.76	0.73	0.73	0.72	0.67	0.02	>0.6	0.489	0.200	0.669	NR	-	-
Serine	0.72	0.73	0.70	0.72	0.72	0.70	0.01	>0.9	0.702	0.868	0.910	NR	-	-
Tyrosine	0.40	0.40	0.36	0.40	0.39	0.39	0.01	>0.9	0.908	0.683	0.696	NR	-	-

Abbreviations: SEM, pooled standard error of treatment means; Adj. R^2^, adjusted R square; NR, no relationship.

**Table 8 animals-13-03586-t008:** FA profiles (% of total fatty acids) of the whole body of rockfish fed the experimental diets replacing various levels of fish meal with tuna by-product meal for 56 days.

	Experimental Diets								Orthogonal Contrast			Regression		
	Con	TBM_20_	TBM_40_	TBM_60_	TBM_80_	TBM_100_	SEM	*p*-Value	Linear	Quadratic	Cubic	Model	*p*-Value	Adj. R^2^
C14:0	2.39 ^a^	2.33 ^a^	2.33 ^a^	2.25 ^ab^	2.24 ^ab^	2.06 ^b^	0.03	<0.002	0.001	0.099	0.250	L	<0.001	0.631
C16:0	15.84 ^c^	16.07 ^bc^	16.29 ^b^	16.30 ^b^	16.69 ^a^	16.76 ^a^	0.08	<0.001	0.001	0.861	0.827	L	<0.001	0.860
C18:0	4.58 ^ab^	4.44 ^b^	4.69 ^a^	4.49 ^b^	4.56 ^ab^	4.58 ^ab^	0.02	<0.02	0.715	0.835	0.887	NR	-	-
C20:0	0.22	0.20	0.19	0.24	0.19	0.18	0.01	>0.1	0.188	0.727	0.185	NR	-	-
C22:0	0.47	0.44	0.46	0.50	0.49	0.49	0.01	>0.7	0.255	0.932	0.303	NR	-	-
C24:0	0.34	0.36	0.38	0.37	0.40	0.35	0.01	>0.3	0.219	0.119	0.464	NR	-	-
∑SFA ^1^	23.84 ^b^	23.85 ^b^	24.34 ^ab^	24.16 ^ab^	24.58 ^a^	24.43 ^a^	0.08	<0.003	0.001	0.320	0.362	L	<0.001	0.534
C16:1n-7	4.52 ^a^	4.40 ^ab^	4.39 ^ab^	4.38 ^ab^	4.21 ^b^	4.19 ^b^	0.03	<0.003	0.001	0.831	0.709	L	<0.001	0.654
C18:1n-9	29.12 ^d^	29.39 ^d^	29.36 ^cd^	29.80 ^c^	30.85 ^b^	32.35 ^a^	0.27	<0.001	0.001	0.001	0.005	C	<0.001	0.981
C20:1n-9	2.84 ^bc^	3.06 ^a^	2.83 ^bc^	2.67 ^cd^	2.96 ^ab^	2.54 ^d^	0.04	<0.001	0.001	0.006	0.700	L	<0.03	0.247
C22:1n-9	0.25	0.26	0.30	0.29	0.24	0.26	0.01	>0.4	0.986	0.173	0.421	NR	-	-
C24:1n-9	1.03	1.03	1.07	1.08	1.07	0.98	0.01	>0.06	0.552	0.007	0.105	NR	-	-
∑MUFA ^2^	37.76 ^e^	38.13 ^c^	37.94 ^d^	38.22 ^c^	39.31 ^b^	40.33 ^a^	0.22	<0.001	0.001	0.001	0.001	Q	<0.001	0.951
C18:2n-6	20.37 ^a^	19.80 ^ab^	19.67 ^abc^	19.11 ^bc^	18.79 ^bc^	18.49 ^c^	0.18	<0.003	0.001	0.807	0.973	L	<0.001	0.729
C18:3n-3	2.50	2.23	2.38	2.41	2.39	2.38	0.03	>0.07	0.844	0.240	0.021	NR	-	-
C20:4n-6	1.91	1.99	2.00	2.04	2.05	2.14	0.03	>0.2	0.016	0.926	0.420	NR	-	-
C20:5n-3	4.38 ^a^	4.25 ^a^	4.21 ^a^	3.86 ^ab^	3.59 ^ab^	3.08 ^b^	0.12	<0.002	0.001	0.096	0.831	L	<0.001	0.696
C22:2n-6	0.45	0.43	0.40	0.40	0.40	0.37	0.01	>0.7	0.173	0.786	0.767	NR	-	-
C22:6n-3	6.28	6.27	6.24	6.24	6.20	6.15	0.03	>0.9	0.314	0.776	0.897	NR	-	-
∑n-3 HUFA ^3^	10.66 ^a^	10.52 ^a^	10.45 ^a^	10.10 ^ab^	9.79 ^ab^	9.23 ^b^	0.14	<0.002	0.001	0.105	0.799	L	<0.001	0.689
Unknown	2.51	3.05	2.83	3.57	2.69	2.64		-	-	-	-	-	-	-

Values (means of triplicate) in the same row sharing the same superscript letter are not significantly different (*p >* 0.05). Abbreviations: SEM, pooled standard error of treatment means; Adj. R^2^, adjusted R square; L, linear; Q, quadratic; C, cubic; NR, no relationship. ^1^ ∑SFA: total content of saturated fatty acids. ^2^ ∑MUFA: total content of monounsaturated fatty acids. ^3^ ∑n-3 HUFA: total content of n-3 highly unsaturated fatty acids.

**Table 9 animals-13-03586-t009:** Hematological parameters of rockfish fed the experimental diets replacing various levels of fish meal with tuna by-product meal for 56 days.

	Experimental Diets								Orthogonal Contrast			Regression		
	Con	TBM_20_	TBM_40_	TBM_60_	TBM_80_	TBM_100_	SEM	*p*-Value	Linear	Quadratic	Cubic	Model	*p*-Value	Adj. R^2^
Plasma parameters														
AST (U/L)	150.7	148.7	149.3	149.0	149.0	149.3	0.54	>0.9	0.645	0.529	0.713	NR	-	-
ALT (U/L)	24.3	23.7	24.3	24.3	24.0	23.0	0.59	>0.9	0.695	0.721	0.698	NR	-	-
ALP (U/L)	184.3	184.0	185.7	183.7	184.3	185.0	0.53	>0.9	0.854	0.942	0.658	NR	-	-
TB (mg/dL)	1.8	1.5	1.3	1.8	1.1	1.5	0.11	>0.4	0.318	0.590	0.872	NR	-	-
T-CHO (mg/dL)	253.3	249.7	248.3	250.3	247.0	249.3	0.90	>0.5	0.189	0.303	0.761	NR	-	-
TG (mg/dL)	363.3	360.7	360.7	363.3	362.3	364.0	0.67	>0.6	0.457	0.282	0.424	NR	-	-
TP (g/dL)	4.6	4.3	4.5	4.8	4.6	4.3	0.10	>0.6	0.808	0.528	0.184	NR	-	-
ALB (g/dL)	1.2	1.2	1.0	1.4	0.9	1.2	0.07	>0.6	0.983	0.858	0.797	NR	-	-
Serum parameters														
SOD (ng/mL)	2.5	2.7	2.4	2.6	2.4	2.4	0.06	>0.6	0.350	0.505	0.791	NR	-	-
Lysozyme (U/mL)	399.9	356.6	327.2	282.5	337.0	357.6	17.59	>0.5	0.417	0.136	0.865	NR	-	-

Abbreviations: AST, aspartate aminotransferase; ALT, alanine aminotransferase; ALP, alkaline phosphatase; TB, total bilirubin; T-CHO, total cholesterol; TG, triglyceride; TP, total protein; ALB, albumin; SOD, superoxide dismutase; SEM, pooled standard error of treatment means; Adj. R^2^, adjusted R square; NR, no relationship.

**Table 10 animals-13-03586-t010:** Effect of dietary treatments on economic parameters of this study.

	Experimental Diets								Orthogonal Contrast			Regression		
	Con	TBM_20_	TBM_40_	TBM_60_	TBM_80_	TBM_100_	SEM	*p*-Value	Linear	Quadratic	Cubic	Model	*p*-Value	Adj. R^2^
Diet price (USD/kg)	1.65	1.57	1.49	1.40	1.32	1.24	-	-	-	-	-	-	-	-
ECR (USD/kg fish) ^1^	1.57 ^a^	1.47 ^b^	1.40 ^bc^	1.32 ^cd^	1.32 ^cd^	1.25 ^d^	0.03	<0.001	0.001	0.035	0.551	L	<0.001	0.878
EPI (USD/fish) ^2^	0.10 ^a^	0.11 ^a^	0.11 ^a^	0.10 ^b^	0.09 ^bc^	0.09 ^c^	0.01	<0.001	0.001	0.003	0.110	Q	<0.001	0.839

Values (means of triplicate) in the same row sharing the same superscript letter are not significantly different (*p >* 0.05). Abbreviations: SEM, pooled standard error of treatment means; Adj. R^2^, adjusted R square; L, linear; Q, quadratic. ^1^ Economic conversion ratio (ECR, USD/kg fish) = Feed consumption (kg) × diet price (USD/kg)/weight gain of fish (kg/fish). ^2^ Economic profit index (EPI, USD/fish) = Final weight (kg/fish) × fish sale price (USD/kg) − ECR × weight gain (kg/fish).

## Data Availability

The data that support the findings of this study are available from the corresponding author upon reasonable request.

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
