# Peer review of "Substitution Impact of Tuna By-Product Meal for Fish Meal in the Diets of Rockfish (Sebastes schlegeli) on Growth and Feed Availability"

_animals, 2023, doi:10.3390/ani13223586_

Round 1

Reviewer 1 Report

Comments and Suggestions for Authors

I believe this is a strong study that was conducted very well with appropriate methods and analysis of results. I highly commend the authors for the economic analyses that were included that incorporated cost of ingredients and feed along with performance characteristics. In my opinion, this is the strongest part of the manuscript and results and will result in a high impact of the findings. I have a few minor points that need to be addressed prior to acceptance for publication. Were all 18 tanks subject to the same water quality parameter changes that are described in lines 128-135? I see some changes in the parameters and assume with flow-through systems, all tanks and fish experienced the same changes and these were not unevenly observed between treatments? Also, there needs to be some clarification on the water source as it is unclear why sand-filtered seawater was mixed with underground seawater, what is the purpose of this mixing for the water that goes through the system?

While the profiles observed for AA and FA of the FM and TBM are not dramatically different, and the subsequent feed formulations are all very similar, I am curious why free amino acids like taurine were not measured or taken into account. It is known that water soluble compounds such as taurine and betaine are lost through the processing of scraps and pressing of fishmeal from these waste/by-product sources. The authors mention fed attractants briefly, but mention very little specifically that could have cause the observed drop-off in performance. Why was taurine supplementation or measurement not considered and is it possible to measure that in feed, ingredients, and tissues to see if it declines as FM is replaced with TBM? I believe a strong correlation may be observed between water soluble compounds like taurine and betaine and performance, as the profiles of feeds are otherwise too similar to explain the performance drop-off at the higher TBM levels. the majority of the other physiological parameters are not significantly different between treatments and would not be expected to be given the high overall similarity in feed profiles.

Comments on the Quality of English Language

English in the abstract and introduction need a bit more editing than the discussion. While it is relatively easy to decipher some sentences and phrasing, some need for correction exists.

Author Response

Reviewer #1

I believe this is a strong study that was conducted very well with appropriate methods and analysis of results. I highly commend the authors for the economic analyses that were included that incorporated cost of ingredients and feed along with performance characteristics. In my opinion, this is the strongest part of the manuscript and results and will result in a high impact of the findings. I have a few minor points that need to be addressed prior to acceptance for publication. Were all 18 tanks subject to the same water quality parameter changes that are described in lines 128-135? I see some changes in the parameters and assume with flow-through systems, all tanks and fish experienced the same changes and these were not unevenly observed between treatments? Also, there needs to be some clarification on the water source as it is unclear why sand-filtered seawater was mixed with underground seawater, what is the purpose of this mixing for the water that goes through the system?

→Yes, all 18 tanks received the same mixture of sand-filtered seawater and underground seawater at a ratio of 1:1. We believe that water quality is same for all 18 tanks. Water temperature commonly increased during summer season, so we used the mixture of the sand-filtered seawater and underground seawater to minimize water temperature’ fluctuation. High water temperature may cause high stress to rockfish.

While the profiles observed for AA and FA of the FM and TBM are not dramatically different, and the subsequent feed formulations are all very similar, I am curious why free amino acids like taurine were not measured or taken into account. It is known that water soluble compounds such as taurine and betaine are lost through the processing of scraps and pressing of fishmeal from these waste/by-product sources. The authors mention fed attractants briefly, but mention very little specifically that could have cause the observed drop-off in performance. Why was taurine supplementation or measurement not considered and is it possible to measure that in feed, ingredients, and tissues to see if it declines as FM is replaced with TBM? I believe a strong correlation may be observed between water soluble compounds like taurine and betaine and performance, as the profiles of feeds are otherwise too similar to explain the performance drop-off at the higher TBM levels. the majority of the other physiological parameters are not significantly different between treatments and would not be expected to be given the high overall similarity in feed profiles.

→We thank you for your valuable comment and kind suggestion. The reduced attractiveness of feed could be due to the loss of water soluble compounds such as taurine and betaine as you explained. We don’t have the equipment (AA analyser and GC) in our lab, so we were unable to measure free AA in our samples due to limited research fund for this study. We sent out the samples for their analysis to another university. However, we agree with your comment that there would be correlations between water soluble compounds (especially, taurine and betaine) and feed intake of fish. Dietary supplementation of taurine or betaine could improve the attractiveness of feed and feed consumption by fish in several studies (Ismail et al, 2020; Zhang et al., 2018). Therefore, we will consider measuring these free AA in our future studies.  

References

Ismail, T.; Hegazi, E.; Dawood, M. A.; Nassef, E.; Bakr, A.; Paray, B.A.; Van Doan, H., 2020. Using of betaine to replace fish meal with soybean or/and corn gluten meal in Nile tilapia (Oreochromis niloticus) diets: Histomorphology, growth, fatty acid, and glucose-related gene expression traits.  Aquac. Rep.   17, 00376.  https://doi.org/10.1016/j.aqrep.2020.100376.

Zhang, J.; Hu, Y.; Ai, Q.; Mao, P.; Tian, Q.; Zhong, L.; Xiao, T.; Chu, W., 2018. Effect of dietary taurine supplementation on growth performance, digestive enzyme activities and antioxidant status of juvenile black carp (Mylopharyngodon piceus) fed with low fish meal diet.  Aquac. Res.   2018, 49, 3187–3195. https://doi.org/10.1111/are.13783.

We really appreciate for your valuable comments on this manuscript.

From corresponding author

Reviewer 2 Report

Comments and Suggestions for Authors

Introduction:

Is there difference between fishery by-products and fish meal? Strictly speaking, fishery by-product meals including tuna bu-product meal belong to the category of fish meal. line 51-52: This is an arbitrary and discriminatory statement, and the two references here are cited without credibility. line 56-74: Authors should focus on the FM replacement with fishery by-products or tuna by-products rather than their blend of fishery by-products or tuna by-products and plant proteins. Authors emphasized they used rockfish at the early stage as target animals in this study. They should explain clearly what is the difference in size between the present study and previous studies. They need to further explain why they conduct the study.

Results:

Orthogonal polynomial contrasts were used to assess the significance of linear or quadratic models. This applies to describing the response of the dependent variable to dietary TBM levels around all data. Table 2 and Table 3 should be listed in the materials and methods. They can be used for subsequent discussions on the impact of TBM substitution for fish meal on performances.

Discussion:

Why did 20% FM replacement by TBM have highest SGR, but 100% FM replacement with TBM exhibit lowest SGR in the case of identical crude protein and lipid levels as well as similar EAA profiles and EFA profiles in the diets which meet the requirements of rockfish growth? This needs in-depth discussion. In addition, authors spent a lot of time discussing the impact of fishery by-product substitution for fish meal on groth performances of fish. However, there is a lack of elaboration on relationships between growth performances and substitution levels of tuna by-prodution meal for fish meal.

Comments on the Quality of English Language

non

Author Response

Reviewer #2

Introduction:

Is there difference between fishery by-products and fish meal? Strictly speaking, fishery by-product meals including tuna by-product meal belong to the category of fish meal. 

→Fish meal is generally made from wild‐caught, small marine pelagic fish nowadays (Jannathulla et al., 2019). Fishery by-product meals including TBM, belong to FM category as you explained, are not commonly considered as a FM source in the aquaculture industry. Since scrap of tuna can be regarded tuna powder meal, TBM is by-product meal of tuna canning process. In the past, tuna by-products including heads, bones, skin, viscera, and fins were treated as waste of tuna. In addition, TBM is much cheaper (58.2% of FM price) than other FM sources. So, we evaluated the potential of TBM as a substitute for FM in the rockfish feeds.

Jannathulla, R.; Rajaram, V.; Kalanjiam, R.; Ambasankar, K.; Muralidhar, M.; Dayal, J.S. Fishmeal availability in the scenarios of climate change: Inevitability of fishmeal replacement in aquafeeds and approaches for the utilization of plant protein sources. Aquac. Res. 2019, 50, 3493–3506. https://doi.org/10.1111/are.14324.

Line 51-52: This is an arbitrary and discriminatory statement, and the two references here are cited without credibility. 

→Thank you for comment. We revised it.

Line 56-74: Authors should focus on the FM replacement with fishery by-products or tuna by-products rather than their blend of fishery by-products or tuna by-products and plant proteins. 

→It was revised as you suggested (Ln 73-78).

 Authors emphasized they used rockfish at the early stage as target animals in this study. They should explain clearly what is the difference in size between the present study and previous studies. They need to further explain why they conduct the study.

→ Fish size (age) could influence the substitutability of the alternative proteins for FM in fish feed as we described at the end of Introduction (Ln 98-100). Dietary TBM substitutability (75%) for fish meal in the late stage of juvenile (initial weight of 29.5 g) was explained (Ln93-96) That is why we conducted this study to evaluate the substitutability of TBM for FM in the diets of the early-stage (initial weight of 2.3 g) rockfish. We also explained it clearly in Discussion (Ln 368-373) about the differences between the present study (initial weight of 2.3 g) and previous study (initial weight of 29.5 g) (Kim et al., 2018). Similar results showing that substitutability of FM with SBM increased from 10% to 40% as rockfish grew from juvenile (1.2 g) to grower (initial weight of 148.2 g) fish (Lee et al., 2016) (Ln 375-379).

Results: Orthogonal polynomial contrasts were used to assess the significance of linear or quadratic models. This applies to describing the response of the dependent variable to dietary TBM levels around all data. Table 2 and Table 3 should be listed in the materials and methods. They can be used for subsequent discussions on the impact of TBM substitution for fish meal on performances.

→We agreed with this comment in which Tables 2 & 3 should be listed in Materials and Methods the manuscript. So we moved Tables 2 & 3 in Materials and Methods.

Discussion: Why did 20% FM replacement by TBM have highest SGR, but 100% FM replacement with TBM exhibit lowest SGR in the case of identical crude protein and lipid levels as well as similar EAA profiles and EFA profiles in the diets which meet the requirements of rockfish growth? This needs in-depth discussion. In addition, authors spent a lot of time discussing the impact of fishery by-product substitution for fish meal on growth performances of fish. However, there is a lack of elaboration on relationships between growth performances and substitution levels of tuna by-product meal for fish meal.

→The TBM20 diet achieved the highest SGR, but no difference in SGR of fish fed the Con, TBM20, and TBM40 diets were found in this study. Slight, but not significant, improvement in SGR (but no difference in weight gain) may be due to the combined protein sources of FM and TBM, resulting in nutrition-balanced in the TBM20 diet. The TBM20 diet with nutrition-balanced might improve SGR slightly in this study. This is not such an importance in this study. However, in regarding the poorest growth of rockfish fed the TBM100 diet, the reduced growth performance was directly resulted from the reduced feed consumption due to deteriorated feed palatability as we explained in Discussion. The reduced palatability could be one of the reasons for the decreased feed intake of fish fed the TBM100 diet (Ln 387-395).

We really appreciate for your valuable comments on this manuscript.

From corresponding author

Reviewer 3 Report

Comments and Suggestions for Authors

The manuscript addresses an important issue in aquaculture: the use of fishmeal. The article deserves better editing and writing, as it presents interesting results.

Introduction: The introduction requires improved writing, with an emphasis on avoiding local descriptions. It would be more engaging to write for a broader audience rather than focusing solely on local readers (Korea).

Materials and Methods: Why didn't the authors calculate the optimal substitution levels using available models such as the broken line, second-order polynomial, and so on? Additionally, consider adding a figure to illustrate these results. Numerous articles suggest that ANOVA or other means of comparison may not be suitable for determining the best level.

Discussion: In my opinion, the discussion section needs improvement. A significant portion of the text focuses on data comparison.

Comments on the Quality of English Language

The manuscript addresses an important issue in aquaculture: the use of fishmeal. The article deserves better editing and writing, as it presents interesting results.

Introduction: The introduction requires improved writing, with an emphasis on avoiding local descriptions. It would be more engaging to write for a broader audience rather than focusing solely on local readers (Korea).

Materials and Methods: Why didn't the authors calculate the optimal substitution levels using available models such as the broken line, second-order polynomial, and so on? Additionally, consider adding a figure to illustrate these results. Numerous articles suggest that ANOVA or other means of comparison may not be suitable for determining the best level.

Discussion: In my opinion, the discussion section needs improvement. A significant portion of the text focuses on data comparison.

Author Response

Reviewer #3

Comments and Suggestions for Authors

The manuscript addresses an important issue in aquaculture: the use of fishmeal. The article deserves better editing and writing, as it presents interesting results.

Introduction: The introduction requires improved writing, with an emphasis on avoiding local descriptions. It would be more engaging to write for a broader audience rather than focusing solely on local readers (Korea).

→Thank you for your comment. It has been revised as you pointed out.  

Materials and Methods: Why didn't the authors calculate the optimal substitution levels using available models such as the broken line, second-order polynomial, and so on? Additionally, consider adding a figure to illustrate these results. Numerous articles suggest that ANOVA or other means of comparison may not be suitable for determining the best level.

→Thank you for your valuable comment. We calculated dietary optimum substitution level of TBM for FM by using second-order polynomial as you suggested and added figure (Fig. 1) to illustrate these results.

Discussion: In my opinion, the discussion section needs improvement. A significant portion of the text focuses on data comparison.

→Thank you for your suggestion. We revised Discussion as you pointed out.  

We really appreciate for your valuable comments on this manuscript.

From corresponding author

Reviewer 4 Report

Comments and Suggestions for Authors

How does the work presented in this manuscript relate to, and expand on, the presentation of previous work using tuna by-product meal? E.g compared to Kim, et al, 2018. Tuna by-product meal as a dietary protein source replacing fishmeal in juvenile Korean rockfish Sebastes schlegeli.

The only difference I see is using smaller sized fish, however this is not very original and does not provide further insight into this topic.

More interesting would be, what is the difference in manufacturing of tuna by-product meal and conventional fish meal?   What is the quality, would it effect digestibility for example?   Tuna by-product meal is actually fish meal, and there are fish meals on the market with the same composition and interesting would be to see e.g.  why feed intake was reduced?  

In general, the manuscript is relatively poorly prepared and researched. The authors extrapolated  more than justified by own data and some of the references are rather dubious.  The work described is rather limited in scope and the manuscript contains little that is fundamentally new.

Introduction too  long and contains plenty of  irrelevant references, see examples

Line 29 – reference [2] not relevant, better to use the original reference (statistics survey etc)

Line 32 – also here [3] the original reference should be used.

Line 38  - reference [6] not suitable, should be the original reference

Line 45  - reference [9]

Line 51  -  commercially, fish skin [11]  is not really considered an alternative protein

Line 52 – this has changed since 10 year ago [11,12] , needs an update

Line 64 – [15] sentence copied straight from reference

Line 68  - with regards to [17]  where is the original reference?

Materials and Methods

The ingredient  composition in Table 1 is apparently expressed on dry matter basis; but what is the dry matter of the tuna by-product meal when sold?

Tuna fish by-product is actually ‘fish meal’, but locally produced, and thus cheaper, but is the quality the same?

Feed intake seems to be reduced, but why?

How was the amount of feed consumed assessed? Do the authors have measures of feed intake or only feed provided; if the latter then the reported values of FER, PER and PPV will not be accurate, and are only crude estimates

Was FER corrected for mortality for example?

What is the rationale of assessing amino acid composition of fish in relation to diet? Amino acid pattern of a protein is typical for a certain species - plant or animal- and does not change regardless of  amino acid pattern of the dietary source.

Results

Body composition of initial sample is missing. How can Protein retention efficiency be calculated ? 

Discussion

The discussion lacks structure, contains many comparisons to, in my opinion, hardly relevant other

studies, and fails to address the value of the study for aquaculture in general; references are taken out of context,  interpretation and synthesis of available information are not very solid and remain very shallow

Line 337  “This disparity could be resulted from that in fish size (initial weight of 2.4 g in this study vs initial weight of 29 g in Kim et al. [21]’s study)”.  It is more likely that increase in weight from 29g to only 53g was too little to see any difference of dietary treatment  

Line 353 “Alanine and serine frequently acted as feeding stimulus for several fish species” the cited references refer to free amino acids which is not measured in this study,

Line 360 to 375  as the amino acid profile was similar in all diets, this discussion about deficiency of AA is futile in the context of this study

Line 374  - as methionine was apparently low in all diets, one cannot claim that growth performance was not negatively influenced if no comparison of growth performance on a higher methionine inclusion is offered; thus this discussion is meaningless.

Line 414 “A diet with high lipid or carbohydrate content could increase liver weight, leading to a higher HSI of fish;  However, no differences in CF, HSI, and VSI of rockfish were found among dietary treatments” this remark is also pointless in the discussion, as diets were isonitrogenous and contained the same amount of lipids.

Line 425  it is well known that whole body protein content does not change drastically and especially amino acid profile is constant.

Line 427 “However, the proximate composition of olive flounder [19] and spotted rose snapper [43] were altered by FM replacement “ the lipid content changed in [19], but not the protein content!  and in reference [43] no significant difference was detected!

Comments on the Quality of English Language

english is fine 

Author Response

Reviewer #4

How does the work presented in this manuscript relate to, and expand on, the presentation of previous work using tuna by-product meal? E.g compared to Kim, et al, 2018. Tuna by-product meal as a dietary protein source replacing fishmeal in juvenile Korean rockfish Sebastes schlegeli. The only difference I see is using smaller sized fish, however this is not very original and does not provide further insight into this topic.

→We think that the smaller fish are more susceptible to lack of dietary nutrition than the larger fish, and substitutability of the alternative source for FM in diet for the smaller fish is also highly limited than the larger fish. This was also proved in Lee et al. (2016) & Burr et al. (2012)’s studies. However, relatively good growth performance (ca. 500% weight gain) of rockfish was obtained in the 8-week feeding trial in this study, but 100% weight gain of rockfish was observed  in the 12-week feeding trial in Kim et al. (2018)’s study. Relatively low growth rate of rockfish in Kim et al. (2018)’s study might be a reason why substitutability of TBM for fish meal in rockfish feeds was high (up to 75% FM) in Kim et al. (2018)’s study.

More interesting would be, what is the difference in manufacturing of tuna by-product meal and conventional fish meal?   What is the quality, would it effect digestibility for example?   Tuna by-product meal is actually fish meal, and there are fish meals on the market with the same composition and interesting would be to see e.g. why feed intake was reduced?  

→Conventional fish meal is generally made from wild‐caught, small marine pelagic fish nowadays (Jannathulla et al., 2019). TBM are not commonly considered as a FM source in the aquaculture industry. Woojin Feed Ind. Co. Ltd. (Incheon, Korea), which is the major TBM producer in Korea, produces constant and stable quality of TBM with several thousand metric tons every year. In addition, price of TBM is much cheaper (58.2% of FM price) than conventional FM sources. As TBM is composed by-products of tuna, which is generated in tuna canning plant, TBM can be considered a replacer for fish meal in fish feeds. As we explained in Discussion section, feed consumption tended to decrease with dietary increased FM replacements with various alternative sources in fish feeds. 

In general, the manuscript is relatively poorly prepared and researched. The authors extrapolated more than justified by own data and some of the references are rather dubious.  The work described is rather limited in scope and the manuscript contains little that is fundamentally new.

→We have carefully revised the manuscript over and over based on 6 reviewer’s comment to improve readability of the manuscript.   

Introduction too long and contains plenty of irrelevant references, see examples

Line 29 – reference [2] not relevant, better to use the original reference (statistics survey etc)

→The sentence was revised and the original reference was used. (Ln 44)

Line 32 – also here [3] the original reference should be used.

→We added the original reference [1] as you pointed out. (Ln 48)

Line 38 - reference [6] not suitable, should be the original reference

→It was revised as you pointed out.(Ln 54)

Line 45 - reference [9]

→It was revised as you pointed out. (Ln 61)

Line 51 - commercially, fish skin [11] is not really considered an alternative protein

→We partially agree with this comment. However, fish skin (Hwang et al., 2014) was also regarded as fishery by-product (olive flounder skin) in rockfish diets in Hwang et al. (2014)’s study, so we think we’d better cite this study in our manuscript.

Line 52 – this has changed since 10 year ago [11, 12], needs an update

→The sentence has been revised. (Ln 67-69)

Line 64 – [15] sentence copied straight from reference

→It was revised as you pointed out. (Ln 80)

Line 68  - with regards to [17]  where is the original reference?

→Unfortunately, the precise statistical data on annual production of TBM in Korea is not available. So we eliminated this sentence.

Materials and Methods

The ingredient composition in Table 1 is apparently expressed on dry matter basis; but what is the dry matter of the tuna by-product meal when sold?

→Dry matter content of TBM is 95.85% when sold. The chemical composition (crude protein: 62.3%, crude lipid: 9.5%, ash: 18.6%) of TBM was used as dry matter basis in Table 1.

Tuna fish by-product is actually ‘fish meal’, but locally produced, and thus cheaper, but is the quality the same?

→Fish meal is generally made from wild‐caught, small marine pelagic fish nowadays (Jannathulla et al., 2019). TBM are not commonly considered as a FM source in the aquaculture industry because TBM is composed of tuna by-products, which is generated tuna canning process, but not including tuna meal. In the past, tuna by-products including heads, bones, skin, viscera, and fins were treated as waste of tuna. That’s why the TBM is much cheaper (58.2% of FM price) than other FM sources. The quality of TBM is very constant and stable now. Woojin Feed Ind. Co. Ltd. (Incheon, Korea) is the main producer of TBM in Korea. In addition, the major feed producers in Korea are using TBM as the FM replacer without any problem.

Jannathulla, R.; Rajaram, V.; Kalanjiam, R.; Ambasankar, K.; Muralidhar, M.; Dayal, J.S. Fishmeal availability in the scenarios of climate change: Inevitability of fishmeal replacement in aquafeeds and approaches for the utilization of plant protein sources. Aquac. Res. 2019, 50, 3493–3506. https://doi.org/10.1111/are.14324.

Feed intake seems to be reduced, but why?

→Feed intake of rockfish tended to reduce with the increased substitution level of FM with TBM in this study. Reduced palatability of feed may be one of the reason for the reduced feed intake of fish. We discussed it in our “Discussion” part. (Ln 387-398)

How was the amount of feed consumed assessed? Do the authors have measures of feed intake or only feed provided; if the latter then the reported values of FER, PER and PPV will not be accurate, and are only crude estimates

→We measured daily feed supplied as we indicated in the footnotes of Table 5 and 2.4. Biological measurement of fish in 2. Material and Method section in the manuscript. We are unable to collect unfed feed from each tank every day due to limited time. However, we have tried to minimize feed waste throughout the 8-week feeding trial. That’s why we got relatively high FE for all experimental diets.

Was FE corrected for mortality for example?

→FE is all correct. We counted total weight of dead fish in our calculation, but we did not indicate it in equation. Now we added the equation including total weight of dead fish in the revised manuscript. 

What is the rationale of assessing amino acid composition of fish in relation to diet? Amino acid pattern of a protein is typical for a certain species - plant or animal- and does not change regardless of amino acid pattern of the dietary source.

 →When we tried to evaluate substitutability of the alternative source for fish meal in fish feeds in the feeding trial, amino acid (AA) profiles of the experimental diets and whole body composition of fish should be analysed to prove that AA profiles of fish was not negatively influenced by dietary FM replacements with the specific alternative source. However, we agreed to your comment that AA profiles of fish are not changed by dietary FM replacement with several alternatives as we already described in Ai et al. (2006)’s study in Discussion section in the manuscript.  Still there are some studies showing that dietary FM replacements with other alternatives changed AA profiles of fish as well as the followings;  

Kari, Z.A., Kabir, M.A., Dawood, M.A., Razab, M.K.A.A., Ariff, N.S.N.A., Sarkar, T., Pati, S., Edinur, H.S., Mat, K., Ismail, T.A., Wei, L.S. Effect of fish meal substitution with fermented soy pulp on growth performance, digestive enzyme, amino acid profile, and immune-related gene expression of African catfish (Clarias gariepinus). Aquaculture, 2022, 546, 737418.            

Roohani, A.M., Abedian Kenari, A., Fallahi Kapoorchali, M., Borani, M.S., Zoriezahra, S.J., Smiley, A.H., Esmaeili, N., Rombenso, A.N. Effect of spirulina Spirulina platensis as a complementary ingredient to reduce dietary fish meal on the growth performance, whole‐body composition, fatty acid and amino acid profiles, and pigmentation of Caspian brown trout (Salmo trutta caspius) juveniles. Aquac. Nutri.  2019, 25, 633–645.

Mastoraki, M., Ferrándiz, P.M., Vardali, S.C., Kontodimas, D.C., Kotzamanis, Y.P., Gasco, L., Chatzifotis, S., Antonopoulou, E. A comparative study on the effect of fish meal substitution with three different insect meals on growth, body composition and metabolism of European sea bass (Dicentrarchus labrax L.). Aquaculture 2020, 528, 735511.

Results

Body composition of initial sample is missing. How can Protein retention efficiency be calculated? 

 →We measured the body chemical composition of initial 10 rockfish samples as we already described in 2.5. Biochemical Composition of the Experimental Diets and Rockfish in the manuscript. We used these data to calculate the protein retention (PR) of fish in this study. Anyway, the crude protein content of the initial 10 rockfish was 15.2%.

Discussion

The discussion lacks structure, contains many comparisons to, in my opinion, hardly relevant other studies, and fails to address the value of the study for aquaculture in general; references are taken out of context, interpretation and synthesis of available information are not very solid and remain very shallow.

→We have carefully revised the manuscript over and over based on 6 reviewer’s comment to improve readability of the manuscript.   

Line 337  “This disparity could be resulted from that in fish size (initial weight of 2.4 g in this study vs initial weight of 29 g in Kim et al. [21]’s study)”.  It is more likely that increase in weight from 29g to only 53g was too little to see any difference of dietary treatment  

→We partially agreed with your comment. However, we think that the smaller fish are more susceptible to lack of dietary nutrition than the larger fish, and substitutability of the alternative source for FM in diet for the smaller fish is also more limited than the larger fish. This was also proved in Lee et al. (2016) & Burr et al. (2012)’s studies. Relatively higher SGR values (2.63–3.018%/day) of rockfish were obtained in the 8-week feeding trial in this study compared to those (0.60–0.69%/day) of rockfish in the 12-week feeding trial in Kim et al. (2018)’s study. Relatively low growth rate of rockfish in Kim et al. (2018)’s study could be another reason why substitutability of TBM for fish meal in rockfish feeds in Kim et al. (2018)’s study was higher (75% FM substitution) than this study (40% FM substitution). We revised this in Discussion (Ln 382-386).

Line 353 “Alanine and serine frequently acted as feeding stimulus for several fish species” the cited references refer to free amino acids which is not measured in this study,

→This sentence was revised and the proper reference was added. (Ln 392-394)

Line 360 to 375  as the amino acid profile was similar in all diets, this discussion about deficiency of AA is futile in the context of this study

→From Line 360 to Line 375, we discussed the importance of balanced amino acid profiles in the experimental diets of fish and whether amino acid profiles of all experimental diets fulfilled the requirement for rockfish in this study. The lysine requirement was met in all experimental diets, and the methionine content in all experimental diets were relatively lower than the requirement for rockfish; we do not think it’s futile to discuss about it.

Line 374  - as methionine was apparently low in all diets, one cannot claim that growth performance was not negatively influenced if no comparison of growth performance on a higher methionine inclusion is offered; thus this discussion is meaningless.

→All experimental diets including Con diet contained lower methionine content than its dietary requirement for rockfish (Yan et al., 2007)’ study, but growth performance of rockfish was quite good in this study. So we’d like to describe to say so. However, this sentence was revised for reader’s readability. (Ln 414)

Line 414 “A diet with high lipid or carbohydrate content could increase liver weight, leading to a higher HSI of fish;  However, no differences in CF, HSI, and VSI of rockfish were found among dietary treatments” this remark is also pointless in the discussion, as diets were isonitrogenous and contained the same amount of lipids.

→It was eliminated as you pointed out.

Line 425  it is well known that whole body protein content does not change drastically and especially amino acid profile is constant.

→Although the whole body composition and amino acid profiles of fish did not change drastically, those parameters still could be changed by dietary FM replacements. Several studies reported that the biochemical composition and AA profiles of the whole body fish were influenced by dietary FM replacements with alternative sources.

Mastoraki, M., Ferrándiz, P.M., Vardali, S.C., Kontodimas, D.C., Kotzamanis, Y.P., Gasco, L., Chatzifotis, S., Antonopoulou, E. A comparative study on the effect of fish meal substitution with three different insect meals on growth, body composition and metabolism of European sea bass (Dicentrarchus labrax L.). Aquaculture 2020, 528, 735511. https://doi.org/10.1016/j.aquaculture.2020.735511.

Arriaga-Hernández, D., Hernández, C., Martínez-Montaño, E., Ibarra-Castro, L., Lizárraga-Velázquez, E., Leyva-López, N., Chávez-Sánchez, M.C. Fish meal replacement by soybean products in aquaculture feeds for white snook, Centropomus viridis: Effect on growth, diet digestibility, and digestive capacity. Aquaculture, 2021, 530, 735823. https://doi.org/10.1016/j.aquaculture.2020.735823.

Wu, Y., Ma, H., Wang, X., Ren, X. Taurine supplementation increases the potential of fishmeal replacement by soybean meal in diets for largemouth bass Micropterus salmoidesAquacul. Nutri, 2021, 27(3), 691–699. https://doi.org/10.1111/anu.13215.

Line 427 “However, the proximate composition of olive flounder [19] and spotted rose snapper [43] were altered by FM replacement “ the lipid content changed in [19], but not the protein content!  and in reference [43] no significant difference was detected!

→We have revised and corrected them as you pointed out. (Ln 465-467)

We really appreciate for your valuable comments on this manuscript.

From corresponding author

Reviewer 5 Report

Comments and Suggestions for Authors

The manuscript “Substitution impact of tuna by-product meal for fish meal in the diets of rockfish (Sebastes schlegeli) on growth and feed availability” by Ran Li and Sung Hwoan Cho describes experimental results on feeding the rockfish, a commercially valuable species in Korean aquaculture, with tuna-based components (0-100% substitution of raw fish with tuna by-product meal). It was found that substantial substitution of the protein source did not to deteriorate growth rate and can be recommended as a profitable protein and mineral source for commercial diets.

Overall opinion

The manuscript fairly corresponds to the journal topics. The content could be of interest to both fundamentals and practitioners (fish farmers, feed producers). Some corrections need to be made for full correspondence of scientific article standards.

Minor concerns

Abstract. I suggest revising the abstract to exclude some redundancy in experimentation details (such as feeding mode, number of individuals, etc.) and to expand the description of findings (results); please assess the relevance of the experiment duration (56 days) mentioned twice or the necessity of the abbreviations used for groups (indicated below, in the main text) in the Abstract, or in opposite decipher the others (e.g. FM).

Line 11 (here or below, in the Materials section), Please specify if the “Fifty-five percent FM” is commonly presented in the rockfish meal?

Introduction, the first phrase. Please revise the double indication on Republic of Korea within the phrase. Please avoid redundant abbreviation, such as “MT” for metric tons which is not referred below in MS.

Table 1. Do you have an explanation as to why the dietary ingredients besides fish meal also varied (fish oil or nutrient composition, for instance)?  

Lines 131-133 It may be better to use “varied” rather than “changed”.

Line 140, Please explain why rockfish survival was estimated on the day of slaughter rather than throughout the experiment; did you mean cumulative total?

Subsection 2.6. The protocols of blood separation described in the first and the second paragraphs are quite similar except for the type of syringe and can be reported together, without redundant words.

Conclusion. I feel that some fundamental results of the research are missed (only a practical summary is given). Please expand the final section by adding the meaningful information.

Technical errors

Please check the text for typos.

Author Response

Reviewer #5

The manuscript “Substitution impact of tuna by-product meal for fish meal in the diets of rockfish (Sebastes schlegeli) on growth and feed availability” by Ran Li and Sung Hwoan Cho describes experimental results on feeding the rockfish, a commercially valuable species in Korean aquaculture, with tuna-based components (0-100% substitution of raw fish with tuna by-product meal). It was found that substantial substitution of the protein source did not to deteriorate growth rate and can be recommended as a profitable protein and mineral source for commercial diets.

Overall opinion

The manuscript fairly corresponds to the journal topics. The content could be of interest to both fundamentals and practitioners (fish farmers, feed producers). Some corrections need to be made for full correspondence of scientific article standards.

Minor concerns

Abstract. I suggest revising the abstract to exclude some redundancy in experimentation details (such as feeding mode, number of individuals, etc.) and to expand the description of findings (results);

→It was revised as you pointed out.

please assess the relevance of the experiment duration (56 days) mentioned twice or the necessity of the abbreviations used for groups (indicated below, in the main text) in the Abstract, or in opposite decipher the others (e.g. FM).

→It was revised as you suggested.

Line 11 (here or below, in the Materials section), Please specify if the “Fifty-five percent FM” is commonly presented in the rockfish meal?

→Amount of 50-55% FM is commonly included in commercial feed for rockfish.

Introduction, the first phrase. Please revise the double indication on Republic of Korea within the phrase. Please avoid redundant abbreviation, such as “MT” for metric tons which is not referred below in MS.

→We revised the double indication about the Republic of Korea as you suggested. In addition, we would not use abbreviation of metric tons although MT (metric tons) was used in Introduction of the manuscript again.

Table 1. Do you have an explanation as to why the dietary ingredients besides fish meal also varied (fish oil or nutrient composition, for instance)?  

→As FM and TBM have different crude protein and crude lipid contents, we have to adjust other ingredients (wheat flour and fish oil) to maintain isonitrogenous (51.0%) and isolipidic (12.5%) in the experimental diets. We revised it as you pointed out (Ln110-111).

Lines 131-133 It may be better to use “varied” rather than “changed”.

→It was revised as you suggested. (Ln 151-157)

Line 140, Please explain why rockfish survival was estimated on the day of slaughter rather than throughout the experiment; did you mean cumulative total?

→Daily survival of fish was monitored when dead fish were found and then immediately removed. The reason we counted total number of fish from each tank is to double-check if daily survival is correct as total number survival fish at the end of the 8-week feeding trial. Daily survival rate and total number of live fish at the end of the 8-week feeding trial are the same and correct in this study.

Subsection 2.6. The protocols of blood separation described in the first and the second paragraphs are quite similar except for the type of syringe and can be reported together, without redundant words.

→Blood of fish was drawn by using heparinized syringe, separated, and used for plasma measurements, such as AST, ALT, ALP, TB, T-Cho, TG, TP, and ALB, while blood of fish were drawn by syringe, separated, and used for serum measurements of fish, such as SOD and lysozyme activity. It was revised (Ln190-204) as you suggested.

Conclusion. I feel that some fundamental results of the research are missed (only a practical summary is given). Please expand the final section by adding the meaningful information.

→It was revised as you pointed out.

Technical errors, Please check the text for typos.

→Thank you for your kind correction. We’ve checked it throughout the manuscript.

We really appreciate for your valuable comments on this manuscript.

From corresponding author

Reviewer 6 Report

Comments and Suggestions for Authors

The study was done for 56 days in triplicate. There was a control batch fed with feed based on fish meal and five substitution variants from 20 to 100%. The chemical composition of the feed, water parameters, biometric parameters of growth, biochemical composition of the meat, hematological parameters and an economic analysis of the use of the feed were determined.

It was concluded that the substitution of fish meal with a percentage between 40-60% is possible.

The study is up-to-date and presents valuable information regarding the possibility of replacing fish meal.

The introduction presents quite brief information about tuna by product and the possibility that it can be a substitute for fish meal.

The monitored water parameters are few (oxygen, salinity, ph and temperature), especially knowing that the use of by-products that are not dried can determine its quality. Ammonia, nitrates and nitrites should also have been monitored. As far as possible, add the other parameters of the water.

Page 114-116 It is not clear how the fodder was made. Were the ingredients put in a blender, mixed with water and then extruded/pelletized? Indicate the extruder/pelletizer model and what temperature is reached during the extrusion/pelletization process. After this process, do the pellets have to be stored cold? Present and developed this aspect.

I consider the rest of the analyzes to be relevant, presented succinctly.

Please develop a little more the part of the conclusions with references to the health of the fish and the quality of their meat

Author Response

Reviewer #6

The study was done for 56 days in triplicate. There was a control batch fed with feed based on fish meal and five substitution variants from 20 to 100%. The chemical composition of the feed, water parameters, biometric parameters of growth, biochemical composition of the meat, hematological parameters and an economic analysis of the use of the feed were determined. It was concluded that the substitution of fish meal with a percentage between 40-60% is possible. The study is up-to-date and presents valuable information regarding the possibility of replacing fish meal.

The introduction presents quite brief information about tuna by product and the possibility that it can be a substitute for fish meal.

The monitored water parameters are few (oxygen, salinity, ph and temperature), especially knowing that the use of by-products that are not dried can determine its quality. Ammonia, nitrates and nitrites should also have been monitored. As far as possible, add the other parameters of the water.

→Thank you for your kind suggestion, however, we did not monitor ammonia products in our study. As we ran this feeding trial in flow-through tank systems with seawater, we never worried about ammonia products in tanks. We ran this system for long time without any ammonia problems in water quality. However, as long as the feeding trial run in the recirculating system or flow-through system with freshwater, we definitely need to monitor ammonia products in water. Those parameters we described are we monitored, so we could not add more in this manuscript. However, we will consider it in future study.

Page 114-116 It is not clear how the fodder was made. Were the ingredients put in a blender, mixed with water and then extruded/pelletized? Indicate the extruder/pelletizer model and what temperature is reached during the extrusion/pelletization process. After this process, do the pellets have to be stored cold? Present and developed this aspect.

→These information was included as you suggested (Ln 133-137).

I consider the rest of the analyzes to be relevant, presented succinctly. Please develop a little more the part of the conclusions with references to the health of the fish and the quality of their meat.

→It was revised and corrected as you suggested (Ln516-520).

We really appreciate for your valuable comments on this manuscript.

From corresponding author

Round 2

Reviewer 2 Report

Comments and Suggestions for Authors

Figure 1: The graphs showed the suitable values for FM replacement by TBM were 2.2% and 7.7% respectively according to the fitting equations, which is a very low FM replacement level compared with the results using ANOVA analysis. Please consider that issue. I suggest that the graphs be used only for showing the declining trends of growth with increasing FM replacement levels by TBM. Accordingly, discussing the part should also be changed.

line 516-520: The appearance of the part is quite abrupt, because these biochemical indicators have already been discussed in the relevant parts of the discussion section.

Poor conclusion with limited useful information.

line 523-524: In fact, high TBM substituting level for FM has adverse effects on growth performance.

line 526-529: a repetitive description.

Comments on the Quality of English Language

The improvement of English language is needed.

Author Response

Reviewer 2

Figure 1: The graphs showed the suitable values for FM replacement by TBM were 2.2% and 7.7% respectively according to the fitting equations, which is a very low FM replacement level compared with the results using ANOVA analysis. Please consider that issue. I suggest that the graphs be used only for showing the declining trends of growth with increasing FM replacement levels by TBM. Accordingly, discussing the part should also be changed.

→Thank you for your kind suggestion. We eliminated dietary optimum substitution level of FM with TBM in Fig. 1 and revised Discussion part accordingly as you suggested. 

line 516-520: The appearance of the part is quite abrupt, because these biochemical indicators have already been discussed in the relevant parts of the discussion section.

→The first sentence was eliminated. However, the second sentence was kept because another reviewer (Reviewer # 6) recommended us developing a little more about the health of fish and the quality of their meat in discussion part. (Ln 513-515)

Poor conclusion with limited useful information.

→The conclusion part was revised. (Ln518-520)

line 523-524: In fact, high TBM substituting level for FM has adverse effects on growth performance.

→The conclusion part was revised. (Ln518-520)

line 526-529: a repetitive description.

→The conclusion part was revised as you pointed out. (Ln518-520)

We really appreciate for your valuable comments on this manuscript.

From corresponding author

Reviewer 3 Report

Comments and Suggestions for Authors

The manuscript has been improved compared with the older version.

However, what calls more attention is that the polynomial regression presents

a result that is totally different from the ANOVA.

A second-order polynomial may not be the best model for describing

and obtaining the substitution level. This point needs to be addressed since a

statistical model indicates a substitution level, and the authors conclude based

on the ANOVA result with a completely different level obtained.

Author Response

Reviewer #3

The manuscript has been improved compared with the older version.

However, what calls more attention is that the polynomial regression presents a result that is totally different from the ANOVA. A second-order polynomial may not be the best model for describing and obtaining the substitution level. This point needs to be addressed since a statistical model indicates a substitution level, and the authors conclude based on the ANOVA result with a completely different level obtained.

→Based on the regression analysis, the best fitting model showed quadratic relationships between dietary substitution level of FM with TBM and weight gain and SGR of rockfish as we described. We agreed to your comment that dietary optimum substitution levels were quite different from the results of their multiple comparison in Table 4. So, we eliminated dietary optimum substitution levels of FM with TBM in the Figure 1. We revised rest of the manuscript accordingly.

We really appreciate for your valuable comments on this manuscript.

From corresponding author

Reviewer 4 Report

Comments and Suggestions for Authors

ND

Author Response

Reviewer 4 

We really appreciate for your valuable comments on this manuscript.